# Effects of 3D Thermal Radiation on the Development of a Shallow Cumulus Cloud Field

Carolin Klinger[1,2], Bernhard Mayer[1], Fabian Jakub[1], Tobias Zinner[1], Seung-Bu Park[3], and Pierre Gentine[3]

[1]Ludwig Maximilians-Universität München, Fakultät für Physik, Meteorologisches Institut München
[2]currently at: Chemical Sciences Division, NOAA Earth System Research Laboratory (ESRL), Boulder, Colorado, USA
[3]Department of Earth and Environmental Engineering, Earth Institute, Columbia University, New York, New York

*Correspondence to:* Carolin Klinger (carolin.klinger@physik.lmu.de)

**Abstract.** We investigate the effects of thermal radiation on cloud development in large-eddy simulations with the UCLA-LES model. We investigate single convective clouds (driven by a warm bubble) at 50 m horizontal resolution and a large cumulus cloud field at 50 m and 100 m horizontal resolution. We compare the newly developed 3D "Neighboring Column Approximation" with the independent column approximation and a simulation without radiation and their respective impact on clouds.

Thermal radiation causes strong local cooling at cloud tops accompanied by a modest warming at the cloud bottom, and in the case of the 3D scheme, also cloud side cooling. 3D thermal radiation causes systematically larger cooling when averaged over the model domain. In order to investigate the effects of local cooling on the clouds and to separate these local effects from a systematically larger cooling effect in the modeling domain, we apply the radiative transfer solutions in different ways. The direct effect of heating and cooling at the clouds is applied (local thermal radiation) in a first simulation. Furthermore, a horizontal average of the 1D and 3D radiation in each layer is used to study the effect of local cloud radiation as opposed to the domain averaged effect. These averaged radiation simulations exhibit a cooling profile with stronger cooling in the cloudy layers. In a final setup, we replace the radiation simulation by a uniform cooling of 2.6 K/d. To focus on the radiation effects themselves and to avoid possible feedbacks, we fixed surface fluxes of latent and sensible heat and omitted the formation of rain in our simulations.

Local thermal radiation changes cloud circulation in the single cloud simulations as well in the shallow cumulus cloud field, by causing stronger updrafts and stronger subsiding shells. In our cumulus cloud field simulation we find that local radiation enhances the circulation compared to the averaged radiation applications. In addition we find that thermal radiation triggers the organization of clouds in two different ways. First, local interactive radiation leads to the formation of cell structures; later on, larger clouds develop. Comparing the effects of 3D and 1D thermal radiation, we find that organization effects of 3D local thermal radiation are usually stronger than the 1D counterpart. Horizontally averaged radiation causes more and deeper clouds then a no radiation simulation, but in general less organized clouds than in the local radiation simulations. Applying a constant cooling to the simulations leads to a similar development of the cloud field as in the case of averaged radiation, but less water condenses overall in the simulation. Generally, clouds contain more liquid water if radiation is accounted for. Furthermore, thermal radiation enhances turbulence and mixing as well as the size and lifetime of clouds. Local thermal radiation produces larger clouds with longer lifetimes.

The cloud field in the 100 m and 50 m resolution simulations develop similarly, however 3D local effects are stronger in the 100 m simulations which might indicate a limit of our 3D radiation parameterization.

# 1 Introduction

Clouds are a key element for accurate climate and weather prediction and cause large uncertainties in the prediction of both (Boucher et al., 2013). Clouds play an important, yet poorly quantified role in climate change. Key questions arising from the limited understanding of clouds in climate prediction were recently pointed out by Bony et al. (2015). These questions include the role of cloud organization or the role of cloud convection in a changing climate. Feedbacks of radiative effects on cloud dynamics and microphysics are one component which modify cloud development and limited understanding of these processes contributes to the uncertainty in climate prediction.

Solar and thermal radiation drive weather and climate and affect cloud formation. Different studies in the past looked at radiative effects caused at clouds. Thermal cooling rates in stratiform clouds were found to vary between 2-40 K/h (Ginzburg, 1984; Davies and Alves, 1989; Stephens, 1978; Feigelson, 1973). These studies showed that thermal heating and cooling rates depend e.g. on the liquid water content of a cloud (Ginzburg (1984), Davies and Alves (1989), Feigelson (1973), Lábó and Geresdi (2016)) and on the strength of inversion layer above the clouds (Twomey, 1983). Harshvardhan et al. (1981) was one of the first looking at 3D effects on clouds by applying radiative transfer to a cuboid cloud. They found strong cooling on the sides of the cuboid cloud, causing a factor up to 3 in the cooling rate when comparing their result to a plane-parallel treatment of a cloud. In a follow up study, Harshvardhan and Weinman (1982) extended this approach to a regular array of cuboid clouds. Again, it was found that 3D cooling rates can exceed those of 1D radiative transfer calculations by a factor of 2-3. Depending on the aspect ratio of the cuboid cloud, this factor can be lower or higher. Guan et al. (1995) further investigated those effects with an axially symmetric 3D radiative transfer model focusing on peaks and holes of isothermal stratiform clouds as well as non-isothermal isolated cylindrical clouds. In holes of stratiform clouds, cooling rates were found to be smaller in 3D compared to 1D radiation simulations. The cylindrical clouds showed cooling at cloud top (up to -34 K/h) and cloud side (-14 K/h). Most of the cooling was found within the first 20 m from the cloud side.

More recent studies using accurate radiative transfer models (e.g. Monte Carlo Models) found strong 3D local thermal cooling rates reaching up to 300-600 K/d (e.g. Kablick et al. (2011); Klinger and Mayer (2014)) in realistic 3D cloud field simulations. It was shown that 3D cooling rates exceed 1D cooling rates both in magnitude and by an additional cloud side cooling. An example of 3D thermal cooling rates in a cumulus cloud field (calculated on an LES time snapshot from Cahalan et al. (2005)) is shown in Figure 1. The figure shows cooling rates at cloud tops and cloud sides, reaching values up to 300 K/d. In addition, modest warming at the cloud bottom (maximum 30 K/d) is found. The resulting change in the surface flux between cloud free and cloudy areas is small in the thermal spectral range. The magnitude of these cooling rates suggests that thermal radiation likely has an impact on cloud development (Davies and Alves, 1989).

Solar radiative effects are different from thermal radiative effects. In the solar spectral range, absorption of sunlight at the

illuminated cloud sides causes heating rates up to 100 K/d (O'Hirok and Gautier, 2005; Jakub and Mayer, 2015a). The location of these heating rates varies with the solar zenith angle. In addition the surface fluxes vary dramatically between directly illuminated and shaded areas (Wapler and Mayer, 2008; Wißmeier and Buras, 2012; Jakub and Mayer, 2015a). Again, the location of the shadow or the directly illuminated surface depends on the solar zenith angle.

The above mentioned radiative effects are known to affect cloud development. Koračin et al. (1998) coupled a 3D Monte Carlo Model to a 2D cloud model. They found situations where 3D solar radiative transfer might become important, but did not investigate the solar effects on cloud development any further. Schumann et al. (2002) studied the cloud shadow effect in the solar spectral range in the convective boundary layer with an idealized setup. They showed non-steady convective motion if the shadow of the cloud was located directly below the cloud and an reduction in the cloud own buoyancy, compared to a shifted shadow. Cloud shading from the anvils of thunderstorm clouds and the potential feedback on thunderstorm dynamics was investigated in a number of studies. Markowski and Harrington (2005) used a very simplified radiative transfer approach by applying a surface cooling of 6 K/h in their simulations. The change in surface sensible heat flux led to differences in the strength of the thunderstorm. The effect of cloud shading and the resulting change in surface fluxes was further adressed by Frame et al. (2009) and Frame and Markowski (2010, 2013).

Thermal cooling is known to drive the development of stratus clouds. Möller (1951) already stated that cooling at cloud top and warming at the cloud bottom can drive convection in clouds by vertical differential heating and cooling and the following destabilization. Therefore stratus clouds might alter to stratocumulus and altostratus to altocumulus. Similar to Möller (1951), Curry and Herman (1985) found increased convection due to radiation. Furthermore they found an increased liquid water content and enhanced droplet growth. Destabilization and enhanced turbulence caused by the vertical differential heating and cooling was found in addition by Sommerai (1976), Fu et al. (1995), Petters et al. (2012) and Lilly (1988). Larson et al. (2001) showed that thermal cooling on the one hand enhances condensation and thus increases liquid water content; on the other hand, radiation causes more entrainment and therefore a decrease in liquid water content. Fu et al. (1995) found that the clear sky cooling enhances convection and increases precipitation by 5%. Tao et al. (1993) showed an increase in precipitation of 14-31% due to thermal effects and Tao et al. (1996) saw an increase in relative humidity of the environment, enhanced circulation and microphysical processes.

In a recent study, Bellon and Geoffroy (2016a) investigated the stratocumulus 1D radiative effect in a set of equilibrium simulations. It was found that depending on the sea surface temperature, radiative cooling at cloud top is crucial for the existence of the equilibrium stratocumulus and causes enhanced turbulence by buoyancy production, more entrainment and a deepening of the boundary layer. Based on the results of the first study, Bellon and Geoffroy (2016b) investigated different approximations of the radiative effect, such as column or horizontally averaged radiation. They found that the radiative effect has to be represented in some detail at the cloud top to account for the enhanced turbulence and mixing and therefore determining the existence of the stratocumulus. Other studies also addressed the differences in cloud development caused by local or homogenized cooling in simulations of deep convective clouds. Xu and Randall (1995) found a longer lifetime of clouds when simulating clouds with local radiation. Thermal radiation increased turbulence on short time scales, and on longer time scales the cloud development itself.

35 In a detailed study, Petch and Gray (2001) investigated the robustness of cloud simulations with varying horizontal resolutions (2 km, 1 km, 500 m), domain size, microphysical parameterizations and different radiation schemes. They also compared results from a 2D and 3D cloud model. In terms of resolution, they found a sensitivity on the mass flux. Turbulence caused by radiative effects was better resolved in the 3D cloud model than in the 2D model. A change in domain size caused a shift in time of major convective events. For their radiation experiments, they differentiated between a no radiation, a local 1D radiation

ation and an averaged radiation application in a 2D cloud model. Ice and mass flux increased in the slab averaged radiation application as well as the amount of condensed water. Petch and Gray (2001) related this to destabilization due to the averaged cooling in areas where there should be no or less cooling in the local application, causing an increase in depth and the rate of development of the clouds. Overall, there were some differences between the averaged and the local radiation case, but the larger one was found when comparing the radiation results to the no radiation simulation. Cole et al. (2005) embedded a 2D

cloud model in a general circulation model (GCM) with 4 km horizontal resolution. They used a two-stream approach for solar radiation and an emissivity approach for the longwave spectrum. They applied the 1D radiation both as local and averaged application and investigated the feedback to cloud development within a 6 month simulation. In addition they compared the results to the standard GCM radiation scheme, which produces also an average radiation tendency per GCM grid box. They concluded that local application of radiative effects causes differences in the development of low and high clouds. The way

the slab average radiation is applied (either calculated from the direct simulation of local effects or from the GCM itself) did not change cloud development in a significant way. The difference between the application of local and homogenized radiation was also addressed by Xiao et al. (2014), with a focus on stratocumulus to cumulus transition. Again, an increase in turbulence, due to the destabilization of the clouds by thermal radiation was found. Xiao et al. (2014) state that because they only used a common 1D approximation for the radiation calculation the effects might be larger with 3D radiation.

In addition to the thermal radiative impacts on clouds development on short time scales mentioned so far, studies by Muller and Held (2012) and Muller and Bony (2015) suggest that local thermal radiation is essential to trigger self-aggregation in radiative-convective-equilibrium simulations. Emanuel et al. (2014) found that clear sky thermal cooling is a key component for self-organization.

Studies coupling thermal interactive 3D radiation to cloud resolving models are rare. Guan et al. (1997) investigated 3D thermal

radiation effects in small cumulus clouds, using a 2D cloud model and the axially symmetric 3D radiative transfer model of Guan et al. (1995). Simulations with 3D longwave radiative transfer were compared to a no radiation simulation. An increase in mean and maximum liquid water content was found. In addition, enhanced downward motion at the cloud sides and an increased upward motion in the cloud center developed in the 3D radiation case. At the end of their simulation, Guan et al. (1997) found an acceleration of the dissipation of the cloud with thermal radiation. Mechem et al. (2008) coupled the 3D radiative

transfer model SHDOM (Evans, 1998) to a 2D cloud model. The effects of 3D thermal radiative transfer on stratocumulus and isolated shallow cumulus were studied. The tendencies of the difference between the 1D and 3D radiative transfer calculation were passed to the cloud model. They found an overall effect of thermal radiation on the development of the cloud field in comparison to a no radiation simulations, but the differences between 1D and 3D thermal radiation were small. Interactive radiation promoted deeper clouds, higher liquid water content as well as cooler and dryer surface conditions. In the stratocu-

mulus case, the main difference was a redistribution of the heating rates in the cloud field.

The microphysical aspect mentioned before will not be addressed in this study, however, for the matter of completeness, we will briefly point out what was found in the past: Harrington et al. (2000) and Marquis and Harrington (2005) showed that thermal emission enhanced cloud droplet growth by diffusion. An earlier onset of collision and coalescence of cloud droplets was found by Hartman and Harrington (2005a) and Hartman and Harrington (2005b) when thermal radiation is considered.

Recent studies of Brewster (2015) and de Lozar and Muessle (2016) emphasize the hypothesis that thermal radiation might influence droplet growth significantly and lead to a broadening of the droplet size spectra and thus enhance the formation of precipitation.

Most of the former studies of cloud-radiation interactions where based on 1D radiation assumptions. The few studies using 3D radiative transfer models are limited using 2D cloud resolving models instead of full 3D cloud models. This paper aims to

address the interaction of radiation and clouds, including a comparison of the effects of 1D and 3D thermal radiation. For this purpose, 3D interactive radiation (the "Neighboring Column Approximation", NCA; Klinger and Mayer (2016)) was developed and integrated into the UCLA-LES (Stevens et al., 2005; Stevens, 2007) and a set of idealized simulations was developed, aiming to isolate the effect of 1D and 3D thermal radiation on clouds. The NCA is fast enough to allow for the first time really extensive 3D thermal radiation studies.

In this paper we extend former studies by applying a fully coupled 3D radiative transfer method in a 3D cloud model and compare 1D and 3D thermal radiative effects. Thermal radiative transfer is applied in a *local* and a *horizontally averaged* setup. We start with simulations of single clouds driven by a heat bubble disturbance and by comparing a no radiation simulation to a 1D and 3D local thermal radiation simulation, thus bridging the gap of the previous study of Guan et al. (1997), where only no radiation and 3D thermal radiation simulations were compared. In a second step, we extend our setup to a shallow cumulus cloud

field at 50 and 100 m resolution, thus increasing the resolution and domain size of previous studies and applying 3D radiative transfer in a 3D cloud model. The model and model setup are described in Section 2. The results are presented in Section 3.

## 2  Simulation Setup

The University of California Los Angeles Large Eddy Simulation model (UCLA-LES; Stevens et al. (2005); Stevens (2007)) is

used for our analysis. The model has previously been successfully used to represent various typical cases, including BOMEX (Cheng et al., 2010), RICO (van Zanten et al., 2011) or DYCOMS (Stevens et al., 2005). The standard UCLA-LES includes bulk microphysics for warm clouds (Seifert and Beheng, 2001) and a 1D radiation scheme ($\delta$-four-stream, Liou et al. (1988)). The spectral integration is accounted for with a correlated-k molecular absorption parameterization (Fu and Liou, 1992). In addition the Monte Carlo spectral integration (MCSI; Pincus and Stevens (2009)) is used in this study for the simulation of the

cumulus cloud field to save computational time. The UCLA-LES was adapted for 3D local thermal radiation by implementing the "Neighboring Column Approximation", (NCA; Klinger and Mayer (2016)) for the calculation of 3D thermal heating and cooling rates.

Two passive scalar tracers were implemented into UCLA-LES, following Park et al. (2016). With the help of the tracers, we performed an octant analysis (Park et al., 2016) to extract coherent structures in simulation data. For further analysis of the results, we used the cloud tracking algorithm Cb-TRAM (Zinner et al., 2008).

## 2.1 UCLA-LES Setup

Two different types of idealized cloud studies have been performed, with either a single cloud or a complete non-precipitating shallow cumulus cloud field.

### Single Cloud

A single cloud, induced by a heat bubble is investigated to study the effects of thermal radiation on individual clouds. We compare the effects of a simulation without radiation (*No Radiation*) to simulations with 1D independent column approximation (*1D Thermal ICA*) and simulations with 3D thermal radiation (*3D Thermal NCA*) using the NCA. For the simulation, the full thermal spectrum was simulated.

As the strength of the radiation effect on cloud development likely depends on the shape and dynamics of a cloud, we choose four different clouds for our investigation.

- A weakly driven, axially symmetric cloud. The heat bubble is introduced by a elliptical shaped volume of warmer air close to the surface. The temperature perturbation is 0.4 K.

- A weakly driven, non-symmetric cloud. The heat bubble is introduced by a uniform random perturbation varying between 0.0 - 0.8 K in the same elliptical shaped volume as the weakly driven symmetric cloud, giving the same average perturbation of 0.4 K as above. The cloud is comparable in strength to the weakly driven axially symmetric cloud.

- A stronger driven, axially symmetric cloud. The heat bubble is introduced by a elliptical shaped volume of warmer air close to the surface. The temperature perturbation is 0.8 K.

- A stronger driven, non-symmetric cloud. The heat bubble is introduced by a uniform random perturbation varying between 0.0 - 1.6 K in the same elliptical shaped volume as the stronger driven symmetric cloud, giving the same average perturbation of 0.8 K as above. The cloud is comparable in strength to the stronger driven axially symmetric cloud.

A stable background profile was chosen in order to cause only moderate updraft velocities of a few m/s. The simulation is performed over 80 min. More setup details are summarized in Table 1. The random noise in the non-symmetric cloud simulations was initialized with the same random seed in all simulations in order to simulate clouds of similar shape which allows a direct comparison of the development of the clouds. The simulations are performed at 50 m horizontal resolution in a 6.4 x 6.4 km$^2$ domain. A stretching of the vertical grid of 1% was applied, starting at 10 m height.

### Shallow Cumulus Cloud Field

Large scale simulations of a shallow cumulus cloud field in a 50x50 km$^2$ domain with 50 m and 100 m horizontal resolution

| Model Variables | Value |
|---|---|
| Number of Grid Boxes | 128 x 128 |
| Number of z-levels | 70 |
| Resolution | 50 m |
| Vertical Stretching | 1 % |
| Surface Perturbation | 0.8 K / 1.6 K |
| SST | 288 K |
| CCN | $70 \cdot 10^6$ 1/kg |
| Microphysics | warm, no rain |
| Variable Output | every 100 s |
| Surface Type | fixed SST |

**Table 1.** Model setup for the single cloud simulations.

have been performed. The environment was that of a warm ocean surface. All simulations are run for 30 hours at Deutsches Klima Rechenzentrum (DKRZ) in Hamburg on Mistral supercomputer (Intel-Haswell) on 512 cores. We focused on the effects of thermal radiative heating and cooling at the clouds itself. Therefore, surface fluxes of latent (180 W/m$^2$) and sensible heat (18 W/m$^2$) were fixed throughout the simulation. The initial atmospheric profiles for this simulations were taken from Stevens (2007). A stretching of the vertical grid of 1% was applied, starting at 100 m height. We allow for warm microphysics (Seifert

and Beheng, 2001), but omit the development of rain to prevent possible feedbacks that rain might cause (e.g. cold pool dynamics) and rather concentrate on radiative effects. Due to the high computational costs of radiation simulations, we used the Monte Carlo Spectral Integration (MCSI, Pincus and Stevens (2009)) in a version adapted for 3D local radiation described in Jakub and Mayer (2015b). Further details are given in Table 2.

Again, we compare different radiation types (*1D Thermal ICA* and *3D Thermal NCA*). Those are the *local radiation* applica-

tions, where heating and cooling acts locally where it is generated. In addition, we *averaged* the thermal heating and cooling of the *1D Thermal ICA* and *3D Thermal NCA* radiation solution in each time step in each layer (*1D Thermal AVG* and *3D Thermal AVG*). These averaged heating rates are then applied in the entire layer to clear sky and cloudy regions. This allows us to separate the effects of local heating/cooling from the systematically larger cooling that is introduced by thermal radiation. Additionally, we apply a *constant cooling* of 2.6 K/d throughout the simulation in the modeling domain. The magnitude of the

cooling was chosen specifically to be comparable to the cooling introduced by the local radiation simulations. The constant cooling differs from the *averaged radiation* simulations in the profile of the cooling. The *averaged radiation* simulations cause more cooling in the cloudy layers.

All simulations are restarted and analyzed after a 3 h initialization run. Until 3 h, the initial simulation is driven by 1D solar and thermal radiation. From the restart time on, we switch on one of the five thermal radiation application or switch radiation

off, thus skipping the spin-up. At 3 h, the first clouds form in the initial run.

| Model Variables | Value |
|---|---|
| Number of Grid Boxes | 1024 x 1024 / 512 x 512 |
| Number of z-levels | 90 |
| Resolution x,y | 50 m / 100 m |
| Resolution z | 30 m |
| Vertical Stretching | 1 % |
| CCN | $150 \cdot 10^6$ 1/kg |
| Microphysics | warm, no rain |
| Variable Output | every 300 s |
| Latent Heat | prescribed: 180 W/m$^2$ |
| Sensible Heat | prescribed: 18 W/m$^2$ |
| Restart | 10800 s |

**Table 2.** Model input for shallow cumulus cloud field simulations.

## 2.2 Cb-TRAM Cloud Tracking Algorithm

To quantify some statistics on the cloud size, lifetime and number of clouds in the simulations, we use a cloud tracking algorithm to track individual clouds over time. Cb-TRAM was originally setup to work with satellite imagery by (Zinner et al., 2008), but is easily adapted to any other 2D information. Here fields of liquid water path are tracked. Cb-TRAM identifies objects as contiguous areas with a specific common characteristic. We set two thresholds to define a cloud: first, only cloud columns of a liquid water path larger then 20 g/m$^2$ are considered; second, a cloud must consist at least of 16 grid connected boxes. Objects defined this way at one time step are identified in the water path field of the next time step using an optical flow analysis of the liquid water field deformation and a simple object overlap analysis. This way cloud objects are detected and tracked over time, allowing us to estimate cloud size and lifetime distributions.

## 3 Simulations Results

### 3.1 Single Cloud Simulations

Figure 2 provides a first impression of the four different single cloud simulations. The visualization shows the weak symmetric, weak non-symmetric, strong symmetric and strong non-symmetric single cloud simulations at 40 min for the *3D Thermal NCA* simulation. The following section provides a detailed analysis of the development of these single clouds for different radiation setups.

A comparison of the time development of liquid water, vertical velocity and cooling rates is shown in Figure 3 and Figure 4. Liquid water path and cooling rates are sampled for the initial cloud only (*conditionally sampled*). During the simulations,

new clouds form close to the surface (see Fig. 2, which we ignore in our analysis. A running average over 300 s was applied to the time series in order to smooth the results. A gray shaded area covers the first 40 min of the simulations in Figure 3 and Figure 4. During this time, the cloud development is dominated by the heat perturbation of the warm bubble. Updraft vertical velocities are strong in this initial stage. We only expect an significant effect of thermal radiation on cloud development after that initial stage. Summing up the thermal cooling in our simulations over time, we found that 40 min is about the time it takes for the thermal cooling to compensate the original heat perturbation of the bubble. This time period is roughly the same in the strong and weakly forced case, because the stronger forced single clouds contain more liquid water and therefore more thermal cooling.

Both, the symmetric and non-symmetric clouds show the impact of thermal radiation on the cloud development. Focusing on the liquid water path (top row of Figure 3) we can split the cloud development into three stages. Both cloud types show a fast development at the beginning (up to about 20 min, gray shaded area, until first gray line). We refer to this first development of the cloud as the "first stage". During that time, the development is dominated by the heat perturbation at the surface. Both liquid water quantities develop similarly in all simulations, with only little differences due to thermal radiation.

After the first stage, liquid water path decreases and finally, from about 30 min onward the cloud stays rather constant at a certain height and does not rise any further. Updrafts become weaker (top row of Figure 4) and radiation acts more significantly on the cloud. All simulations show that the liquid water path (top row of Figure 3) is reduced in this "second stage" (from about 20 min to 40 min). The "last stage" of cloud development (about 40 min to 80 min) is dominated by a second growth-period of the cloud. In this stage, thermal radiation can act on the cloud. Both non-symmetric clouds show a second rise in liquid water path, in case of the stronger forced cloud exceeding the *No Radiation* simulation. When comparing the development of liquid water path to the development of maximum liquid water mixing ratio over time (lower row of Figure 3) the rise in liquid water mixing ratio in the last phase becomes more evident. Maximum values of liquid water mixing ratio in the radiation simulations exceed the *No Radiation* simulation in this last phase. Looking at the location of these maximum values of liquid water mixing ratio and the shape of the cloud (not shown), we find that clouds become narrower (have a reduced horizontal extent) over time when radiative effects are accounted for and maxima of liquid water mixing ratio are enhanced in the center of the cloud. Vertical velocities show stronger upward and downward values for both thermal radiation simulations. The upward and downward motion in the first stage results from the initial temperature perturbation and the resulting overturning circulation.

Looking at the differences between *1D Thermal ICA* and *3D Thermal NCA* we find different amounts of cooling in the *3D Thermal NCA* radiation simulation which affects the further development of the cloud (bottom row of Figure 4). Differences occur in terms of liquid water when comparing both thermal radiation simulations. The differences are small, but in general slightly stronger in the case of *3D Thermal NCA* simulation. In the last stage, differences in vertical velocity between the *No Radiation* and the *thermal radiation* simulations are evident. Both radiation simulations show stronger upward and downward vertical velocities. Vertical velocities are usually a bit stronger in case of *3D Thermal NCA* radiation than in case of 1D thermal radiation. Combining the development of liquid water with upward and downward vertical velocities (top and middle row of Figure 4), the data suggests that a change in cloud circulation is induced from the second stage onward by the effects of thermal radiation, which enhances updraft vertical velocities in the cloud cores, thus strengthening the cloud development and at the

same time, induces stronger downdraft vertical velocities at the cloud sides.

The region of subsiding motion around a cloud, the "subsiding shell" is known from previous work of Heus and Jonker (2008); Jiang et al. (2006); Small et al. (2009). Heus and Jonker (2008) studied the subsiding shell of cumulus clouds from measurement data in comparison to model simulations and concluded that negative buoyancy at the cloud sides causes the subsiding

shell to develop. Jiang et al. (2006) and Small et al. (2009) compared measurement data and simulations of small cumulus clouds to investigate the effect of different aerosol concentrations on cloud lifetime. They found a stronger subsiding shell due to enhanced evaporative cooling. Stronger downward motion at cloud sides due to thermal radiation was found by Guan et al. (1997).

To further investigate this question and the development of the cloud, with a special focus on the possible change in cloud

circulation and the subsiding shell, Figure 5 shows transects of liquid water (bottom of each figure) and vertical velocity (center of each figure) in the last stage of cloud development. Both quantities displayed are averages of the 900-1200 m height layer. In the time series analysis, the cloud development is accompanied by stronger updrafts and stronger downdrafts in the simulations with radiation. Figure 5 shows the stronger upward and downward motion in the transect. The subsiding shells are clearly visible at the cloud side region. Vertical velocities increase from -0.1 m/s in the *No Radiation* simulation to about

-0.8 to -1.0 m/s in the simulations with radiation, peaking for the *3D thermal radiation* simulation, although the difference between 1D and 3D radiation is modest. Liquid water content is enhanced in the simulations with radiation in the cloud center and the cloud is narrower (Figure 5). This is in agreement with the results of the time development of liquid water path and maximum liquid water (Figure 3) that indicated narrower clouds with enhanced liquid water content in the cloud center.

Finally, Figure 6 shows the horizontally averaged vertical profile of negative and positive buoyancy, sampled in the cloudy

region (all grid boxes where liquid water is larger than zero). Data is sampled for 8 min, starting after 50 min of the simulation. All simulations including thermal radiation show stronger negative buoyancy which is slightly larger in the *3D Thermal NCA* case than in the 1D case. This negative buoyancy is due to the thermal cooling at cloud tops, and in case of *3D Thermal NCA* radiation at cloud sides. The negative buoyancy can cause the observed subsiding motion, as already found by Heus and Jonker (2008) and Small et al. (2009). In addition, the stronger horizontal buoyancy gradient (difference between positive and negative

buoyancy in Fig 6) generates enhanced turbulence and lateral mixing and therefore stronger evaporation. The stronger evaporation explains the narrowing of the clouds in the horizontal seen in Figure 5. Enhanced evaporation can cause an additional cooling and therefore a positive feedback to the already existing horizontal buoyancy gradient. If thermal radiation itself, or the enhanced evaporation contributes more to the formation of the subsiding shell cannot be said from our current simulations. However, it is certain that thermal radiation strengthens the development by generating negative buoyancy.

These results confirm what was found by Guan et al. (1997). They also found an increase in liquid water, both in term of average and maximum values and stronger downward motion at the cloud side. While Guan et al. (1997) compared a no radiation simulation to a 3D thermal radiation simulation of a symmetric cloud, we include 1D radiative transfer in our study and extend the study to non-symmetric clouds. The differences between 1D and 3D thermal radiative transfer are small. 3D thermal radiation causes a slightly stronger effect than 1D thermal radiative transfer though. As the effect of 3D thermal radiation is

the additional cloud side cooling, the magnitude of this cooling determines on how strong the 3D thermal radiation effects

are. The amount of cooling in turn is related to the cloud side area. As Figure. 2 shows, our clouds are rather oblate, which reduces the 3D radiation effect since the side area is small compared to the top area. In addition, the limitations of the NCA as discussed later in this paper (Section 3.2.4) might cause some neglect of the cloud side cooling. We summarize therefore that we can confirm previous findings that thermal radiation changes the cloud circulation and enhances liquid water content with 1D thermal radiation being nearly as efficient as 3D thermal radiation. The magnitude of the 3D effect might depend significantly on the shape of the cloud (cloud type) and requires further study.

## 3.2 Shallow Cumulus Cloud Field Model Experiments

In this section, we explore the effects of thermal radiation on the development and organization of shallow cumulus clouds in the 50 x 50 km$^2$ domain at 50 m horizontal resolution. Figure 7 shows the cloud field for three of the six performed simulations. The figure shows a time snapshot at 20 hours of the simulations. We can already see in this snapshot that clouds organize differntly, depending on the radiation application used.

Figure 8 shows the temporal development of cloud fraction and maximum liquid water mixing ratio from the restart time until 30 hours. In addition to the *local* 3D and 1D *Thermal radiation* cases, the *averaged* and fixed (*constant cooling*) radiation scenarios are shown. In the *No Radiation* simulations, cloud cover stays constant at about 10 - 12 % from 8 hours on with a slight decrease towards the end of the simulation. Maximum liquid water mixing ratio is less in the *No Radiation* simulation, compared to the simulations with radiation (1.9 g/kg versus 3.6 g/kg). Differences are also found between the *local radiation* simulations and the *averaged radiation* simulations. Cloud cover increases more rapidly in the *averaged radiation* simulations compared to the *local* ones. The maximum of liquid water mixing ratio, however, shows an opposite development. It is therefore likely that clouds organize differently, depending on the treatment of radiation. The two gray lines at 10 hours and 20 hours indicate stages where the development of the simulation changes. While all simulations with radiation perform nearly the same until 10 hours, they start to differentiate afterwards. Maximum liquid water mixing ratio exceeds 2 g/kg after 20 hours. In a rain permitting simulation rain would likely form at that time. We ran the simulations for 30 hours, to see what would theoretically happen to the clouds, but as we are aware that the simulations become more and more unrealistic from 20 hours on, our main analysis will be on the time interval between restart and 20 hours. After 20 hours, a strong increase in cloud fraction in all simulations with radiation is found. This increase is particularly strong for the two *averaged radiation* simulations. Liquid water mixing ratio (and the liquid water path, not shown) starts varying more rapidly, indicating stronger formation of clouds and their decay.

Figure 9 shows the temporal development of liquid water path (sampled in the entire domain), maximum vertical velocity and cloud base and top height until 20 hours. Liquid water path increases with time in all simulations. The increase is strongest in the case of *3D Thermal NCA*. Each of the *local thermal radiation* simulations produces more liquid water than its *averaged* counterpart. This differences are small however. The different development of cloud cover and liquid water in the *averaged* and *local thermal radiation* simulations result in smaller differences between these two setups. Liquid water path increases less in the simulation with *constant cooling*. The maximum vertical velocities are weaker in the *No Radiation* simulation and

cloud vertical extent is smaller with a heightened cloud base and lower cloud tops. All quantities increase over time. Due to the missing cooling, less water condenses, the number of clouds, cloud cover and a liquid water path is reduced. The higher cloud base is also a result of the missing cooling which leads to a warmer temperature profile in the *No Radiation* simulation. The smaller increase in cloud top height might be linked to reduced updraft velocities and missing destabilization (due to thermal
cooling) in the *No Radiation* simulation (see Section 3.2.1).

To study whether the observed differences are simply caused by the systematic change introduced by 1D or 3D radiation, or if the local effects are relevant, we compare the *averaged* and *local* thermal radiation simulations. Although the averaged amount of cooling per domain is (in the beginning of the simulation) the same for both *1D Thermal ICA* and *1D Thermal AVG* (or *3D Thermal NCA* and *3D Thermal AVG*) simulations, the *averaged radiation* simulations produce higher cloud cover than
their corresponding 1D or 3D *local radiation* simulation. Liquid water path and maximum liquid water content develop in the opposite direction: both are lower for the *averaged radiation* simulations until 20 hours. The development of all quantities shown here can be related to the location where thermal radiation acts. In the *local radiation* simulations, cooling (and some warming at the cloud bottom) acts directly at the cloud edges. Cooling rates can locally be up to several hundred K/d and can destabilize the cloud layer, thus promoting updrafts, more condensation and an increase in cloud height. For the *averaged*
*radiation* simulations, cooling occurs in the cloud layers, but as it is averaged and applied everywhere in a layer, independent of the location of the clouds, a destabilization occurs in the whole area. However, the destabilization of the *averaged radiation* simulations is weaker at the clouds than the local destabilization of the *local radiation* simulations. Finally, in the constant cooling simulation, the cooling is distributed equally over all heights. Cooling in the cloud layer is thus smaller compared to the simulations with *local* or *averaged* radiation which explains the lower liquid water path.

### 3.2.1   Boundary Layer and Cloud Layer Development

We investigate the development of the boundary layer and cloud layer by examining the profiles of different quantities at three time periods of the simulation. Starting with the initial profile at the restart time (3 hours), we show in addition the averaged profile from 9-11 hours (noted as 10 hours) and from 19-21 hours (noted as 20 hours).
The initial profiles at 3 hours (Figure 10 and Figure 11, first column) show the typical profiles of a boundary layer over a warm ocean surface. No clouds have developed yet at this stage. The profile of liquid water potential temperature shows a well mixed layer (up to 400 m), the conditionally unstable layer (up to 1200 m) as well as the inversion layer at about 1100 m height. Relative humidity increases with height at first, before decreasing from 400 m height to the inversion, indicating the entrainment of dry air from aloft the inversion. Typical for the boundary layer over a warm ocean, turbulence is produced by
buoyancy in the layer close to the warm ocean (rising thermals). The upward motion of this low layer can also be seen in the updraft velocity in the lower layer until 400 m height (Figure 11, middle).

The first clouds appear shortly after the restart in all simulations. From this time on, thermal radiation (that is cloud top cooling and cloud bottom warming, and in the case of *3D Thermal NCA* cloud side cooling) changes the development of the boundary layer and of the clouds themselves. Due to the imposed constant surface fluxes of latent and sensible heat, the atmosphere

warms over time. When thermal radiation is applied in the simulations, this warming is partially compensated by thermal cooling. At 10 hours, the whole atmosphere is about about 1 K cooler in the thermal radiation simulations (see Figure 10) which in turn leads to a higher relative humidity and more condensation of water vapor. The increase in liquid water over time was already shown in Figure 9. Here, in addition to the increased liquid water (which is stronger for the *3D Thermal NCA* radiation case at 10 hours although differences are small) a deepening of the cloud layer occurs in the simulations including thermal radiation. Thermal cooling at the cloud boundary (*local radiation* simulations) and in the cloud layer (*averaged radiation* simulations) cause more condensation. The *constant cooling* simulations produces less water, because the cooling is not directly produced by the clouds but imposed in the simulation setup in the whole atmosphere.

The cooling at cloud tops (and cloud sides in the 3D radiation case) as well as the bottom warming leads to a destabilization of the cloud layer, promoting the development of clouds by increased buoyancy (Figure 11, second column). Turbulence that is initially only produced through surface flux induced buoyancy tendencies is now additionally produced in the cloud layer. Both *local radiation* simulations show more buoyancy production than the *averaged radiation* simulations, again with the *3D Thermal NCA* simulation showing slightly stronger values then the *1D Thermal ICA* simulation. Due to the increased buoyancy in the simulations with radiation, upward velocities in the clouds are stronger (second column of Figure 11, middle). Furthermore, all simulations with radiation produce stronger downdraft vertical velocities in the subsiding shells, especially the *local radiation* simulations (Figure 11, bottom).

The difference in the temperature profiles between the *No Radiation* and the simulations with radiation increases (up to 3 K), which again, leads to an increase in relative humidity in the sub-cloud layer and more condensation. We note here that in the cloud layer, the relative humidity decreases in the *local radiation* simulations, because the liquid water mixing ratio is higher, although the temperature is lower. The production of TKE through buoyancy is shifted upward into the cloud layer and upward velocities increase in the cloud layer, which becomes deeper (see the deepening of liquid water profile, Figure 10). While at the beginning of the simulation, the *3D Thermal NCA* simulations produced the largest amount of liquid water, the averaged radiation simulations produce the largest amount at the end of the simulation. The development of liquid water, relative humidity and the TKE production by buoyancy and the development of vertical velocities suggest that more mixing/entrainment of dry air from aloft the cloud layer occurs in the *local radiation* simulations.

We summarize therefore that *1D Thermal* and *3D Thermal* heating and cooling at clouds destabilizes the cloud layer, promoting the development of strong updraft cores and the transport of water vapor into the cloud layer. In addition, the thermal cooling of the atmosphere leads to enhanced condensation. Mixing in the cloud layer is stronger. In addition to the stronger updraft velocities, downdrafts increase as well.

The local cooling at the cloud boundary itself in the *local radiation* simulations (in comparison to the *averaged radiation* effect), increases the earlier described development by destabilizing the cloud layer locally at the clouds stronger than in the *averaged radiation* simulations. Entrainment is stronger in the *local radiation* simulations, causing less condensation and lower relative humidity. The simulation with *constant cooling* usually shows the weakest effect of all simulations with radiation.

We hypothesize that thermal radiation, and especially the localized thermal heating and cooling (as was already shown for the single cloud simulation) leads to stronger development of the cloud circulation in terms of updrafts and subsiding shells. *3D*

*Thermal NCA* radiation, in comparison to 1D thermal radiation shows a slightly stronger increase of these shown effects by an additional cloud side cooling and overall stronger cooling in the modeling domain.

### 3.2.2 Cloud Development

The preceding section (Sec. 3.2.1) analyzed the effects of thermal radiation on the development of the cloud-topped boundary layer. In this section, we further investigate the effects of thermal radiation on cloud development. It was shown before that the cloud circulation changes due to the effects of thermal radiation, promoting updrafts and subsiding shells, a deepening of the clouds, and depending on the radiation type, increased liquid water within the clouds. Another hypothesis raised earlier is the possible organization of clouds (see beginning of Sec. 3.2) due to thermal radiation. In addition, thermal radiation may alter cloud lifetime. In the following we will address these possible changes.

*Cloud Circulation*

Results from the single cloud simulation and the statistical analysis of the cloud field simulations suggest that a change in the cloud circulation occurs, promoted by thermal radiative heating and cooling at the clouds. Stronger updrafts and stronger downdrafts/subsiding shells are expected due to the destabilization of the cloud layer and by thermal cooling of the clouds. Therefore, changes in the cloud circulation are expected to be stronger for the simulations with local *1D Thermal ICA* and *3D Thermal NCA* radiation, compared to the horizontally *averaged radiation* simulations. All simulations with radiation are expected to show stronger circulation features than the *No Radiation* simulation.

For the cloud field simulations, we used the octant analysis described by Park et al. (2016) to extract updrafts and subsiding shells from our simulations. By the signs of flux perturbations, eight parts (octants) are derived from the spatial field of three variables (vertical velocity and two passive scalars). Those octants include updrafts and subsiding shells/downdrafts (note that the analysis does not separate downdrafts inside clouds and subsiding shells). The analysis is restricted to cloudy layers (layers, where at least one grid box has a liquid water mixing ratio larger than 0.1 g/kg). Figure 12 shows the averaged and maximum updraft and downdraft velocities over time. Updrafts are stronger in all thermal radiation simulations, compared to the *No Radiation* case. The updraft velocities of the *local radiation* simulation are slightly stronger than the updrafts in the *averaged radiation* simulations. The *local radiation* simulations produce stronger subsiding shells, noted in the averaged as well as maximum values. Updrafts and downdrafts in the *3D radiation* cases are in general slightly stronger than their *1D* counterparts. Therefore an overall stronger circulation, induced by local heating and cooling is found. These results agree with the increase in buoyancy production, the development of relative humidity and upward vertical velocity as shown in Sec. 3.2.1. In terms of updrafts and subsiding shells we find the *constant cooling* simulations produces similar results as the *averaged radiation* simulations.

*Cloud Organization*

Apart from the changes in the cloud circulation, clouds organize differently. Cloud cover and liquid water developed differently for the individual simulations. The horizontally *averaged radiation* simulations showed larger cloud cover over time than the *local radiation* simulations.

We investigate the cloud fields at 15 and 20 hours of the simulations (Figure 13). Comparing the *No Radiation* simulation to
the thermal radiation simulation we see smaller, less deep and fewer clouds. As seen before, the simulations with *constant cooling* or *averaged radiation* show a similar behavior. Deeper clouds with higher liquid water content and a higher number of clouds are found here. At 20 hours there seems to be a tendency for patches to from. The *local radiation* simulations show a completely different development. At 15 hours, we can see a separation into cloud free and cloudy areas. The clouds are larger then in the *averaged radiation* simulations. This development continues and leads to cell structures at 20 h, similar to those
usually found due to cold pool dynamics (e.g. Sharon et al. (2006), Xue et al. (2008), Savic-Jovcic and Stevens (2008)). The *3D Thermal NCA* simulation also shows larger clouds at 20 hours.

To investigate how cloud size changes over time, we calculated the temporal variation in the autocorrelation length (defined by the shift where the correlation coefficient drops below 1/e) of the liquid water path of the cloud field (Figure 14). At about 9 hours, the simulations start to develop differently. Both *local radiation* simulations show an increased correlation length from
this time on, indicating larger clouds. The largest clouds are found for the *3D Thermal NCA* simulation, especially towards the end of the period shown here. Both *averaged radiation* simulations behave quite similar and show less organization than the *local radiation* simulations, but slightly more than the *No Radiation* simulation. The *constant cooling* simulation is located between the *averaged radiation* simulations and the *No Radiation* simulation. A small increase in the correlation length is found at the end of the investigated period for the *averaged radiation*, which agrees well with Figure 13.

It shall be mentioned here (although not shown for the above mentioned reasons) that from about 24 hours on, large clouds form in the *averaged radiation* simulations and the *const cooling* simulation, in which the clouds oscillate: disappearing and then reappearing. No systematic difference between 1D and 3D radiation is found in these cases. The *local radiation* simulations still show cells, however, clouds become larger, especially in the *3D Thermal NCA* simulation.

To further investigate how much water the individual clouds contain and if and how they organize, we show Hovmoeller di-
agrams (Figure 15). Liquid water path was averaged in x-direction for these diagrams. They thus provide an overview of the spatial and temporal development of the cloud field. Extended patches of liquid water along the spatial dimensions indicate large clouds. Extended patches along the time axis show long living clouds. In additions, the diagrams show how much water is located in the cloud patches. In the *No Radiation* simulation, no organization occurs. Clouds remain small with little liquid water content throughout the simulation. This is different if thermal radiation is accounted for. If we compare the five different
simulations with radiation, one notices that both *local radiation* simulations and both horizontally *averaged radiation* simulations as well as the *constant cooling* simulation show a similar behavior.

Larger fibers, containing more liquid water content are found earlier in the simulations with *local thermal radiation*. Patches of dry and wet regions form (blue till red vs. white areas). The simulations with the horizontally *averaged radiation* show the first indications of larger structures at the end of the study period, but they contain less liquid water then the *local thermal*
*radiation* simulations. Also, no significant differences exists between the 1D and 3D *averaged radiation* simulations (which

was also evident in Figure 14). Comparing the 1D and 3D *local thermal radiation* simulations, we find at the end of the shown period first larger structures in the *3D Thermal NCA* radiation simulation. The fiber-like structures of the Hovmoeller diagrams usually give a hint on the movement of the clouds. Here, however, the structures show that clouds move very little during their lifetime and mostly remain at one location, as our simulation is performed without any background mean wind.

We can therefore summarize the following findings: Local thermal radiation enhances cloud organization in our simulations in the first 20 hours by forming cell structures and larger clouds, concentrating more liquid water in individual clouds.

*Cloud Lifetime*

The increase in cloud lifetime which could already be seen in the Hovemoeller diagrams is quantified in this section. Figure 16

shows a probability density function (pdf) of cloud lifetime. Each cloud occurring within the first 20 hours of our simulations was tracked and the lifetime was calculated. *Local thermal radiation* leads to less clouds with a small lifetime, but more clouds with a larger lifetime.

The results of cloud size and lifetime agree with the results of the last paragraph concerning the cloud organization. Cloud organization and the size-dependence of cloud lifetime are closely related, as smaller clouds dying and larger clouds growing

will result in fewer but larger clouds and in longer correlation lengths. In addition, enhanced turbulence and mixing in the *local radiation* simulations can lead to a faster decay of small clouds, while larger clouds might live longer and grow due to the enhanced cloud circulation (stronger updrafts and downdrafts). We suspect that similar to the single cloud experiment, *local thermal radiation* reduces cloud diameter and therefore reduces small-size clouds in our simulations. The *averaged radiation* simulations show more clouds with a longer lifetime than the *No Radiation* and the *const. cooling* case.

Figure 17 shows the pdf compared to simulation time and the cloud size. Many small clouds occur at the beginning of the simulations. In the *local radiation* simulations, these small clouds become fewer over time and more larger clouds occur, while in all other simulations, these small clouds occur throughout the whole simulation. Clouds larger than $1 \, 10^3$ km$^2$ occur in the *local radiation* simulations, but hardly any in the *No Radiation* and the *const. cooling* case.

*Summary of Cloud Development* In this section we investigated the effects of different radiation application on the development of a shallow cumulus cloud field. Summarizing, we found that there is a definite difference between a *No Radiation* and thermal radiation application, where thermal radiation causes more condensation, deeper and more clouds and stronger up- and downdrafts. The boundary layer and cloud layer becomes deeper and more mixing from the aloft inversion layer occurs. The *averaged* and the *local radiation* simulations differ in terms of cloud size, number of clouds and the organization of the

cloud field. We find cell structures and larger clouds in the *local radiation* simulation until 20 hours. Simulations with *3D local radiation* develop larger clouds.

In that context it is not relevant for the organization of the cloud field if the averaged radiation was calculated from the 3D or the 1D simulations or if even a prescribed cooling of 2.6 K/d was applied, similar to the findings of Cole et al. (2005).

When we consider *local radiation* the differences between 1D and 3D thermal radiation are also not large, but we see a ten-

dency at the end of our 20 hour time period for the formation of larger clouds. This is a similar result to Mechem et al. (2008)

where differences between 1D and 3D thermal radiative transfer in a shallow cumulus cloud field were small as well. We will address the issue of these small differences between the 1D and 3D *local radiation* simulation in Section 3.2.4 again.

The main difference between the *averaged* and the *local radiation* is the location and the strength of the thermal cooling (as shown in Figure 18 which summarizes the results of this subsection). In the case of *local radiation*, the cooling (or heating) acts

locally at the cloud sides, tops and bottom. Cooling rates can be as large as several 100 K/d. This causes a local destabilizing. This is supported by stronger updrafts and downdrafts for the *local radiation* simulations. The stronger entrainment caused by cloud side cooling shrinks clouds. At the same time, the stronger updrafts lead to stronger cloud growth. It is possible that these two processes vary with the perimeter to area ratio of updrafts, and so the first process can be expected to win for small clouds, and the second one for large clouds.

In case of *averaged radiation*, the resulting cooling is weaker but acts in the entire modeling domain and the cooling does not distinguish between cloudy and cloudless regions. It therefore takes longer for the *averaged radiation* to destabilize the atmosphere where clouds are located. At the same time, clear sky regions are stronger destabilized, promoting new development of clouds. This can explain why more and smaller clouds are found during the first 20 h of the simulations. However, when a certain destabilization is reached, it causes a rapid cloud development in the entire domain at once (in our simulations after 24

hours).

### 3.2.3   Dependence of the Results on Resolution and Reproducibility

One important issue of our simulations is the robustness of the results and the dependence on resolution. We therefore repeated the calculation with a horizontal resolution of 100 m instead of 50 m and performed three runs for the computationally cheaper

100 m resolution. Although some differences occur between the three 100 m simulations (one reason being for example the randomly chosen spectral bands in the MCSI), the effects (e.g. stronger organization or the more locally focused liquid water in the 3D *local radiation* radiation case) remain and are even stronger. This is at first a counterintuitive result, because radiation effects, and also the 3D radiation effects are expected to be stronger the better the model resolution. We will address this issue in the next subsection (Sec. 3.2.4).

We now focus on some aspects of the 100 m resolution simulation. Figure 19 shows the time series of cloud fraction and maximum liquid water mixing ratio. In this figure we also show the results of the additional two simulations (thinner lines). Similar to the 50 m resolution simulations, cloud cover is largest for the *averaged radiation* simulations. The difference between the 1D and 3D radiation simulations, both for *local* and *averaged* radiation are larger than in the 50 m resolution simulations. In terms of liquid water, the same development is found in the 100 m resolution simulation as in the 50 m resolution simulation.

3D effects are also stronger here. Liquid water path and vertical velocity (not shown) show slightly stronger 3D effects in the 100 m resolution simulations then in the 50 m resolution simulations, while the difference in cloud base and height remains the same.

Concerning the organization, we start again with snap shots at 15 and 20 hours of the 100 m resolution simulation, shown in Figure 20. As might be assumed from the time series, differences between 1D and 3D *local radiation* radiation are larger than

in the 50 m resolution simulation. At 15 hours, we find first separation in cloud free regions and regions of deeper clouds with larger clouds in the *3D Thermal NCA* radiation case. The *average radiation* simulations show, similar to the 50 m simulations a rather equal distribution of small cloud. At 20 hours, we find again the cell structures, but only in the *1D Thermal ICA* radiation case. In the *3D Thermal NCA* radiation case, clouds have formed one large patch (if we account for the periodic boundary conditions of the simulation). To account for the whole modeling period, Figures. 21 and 22 show the Hovmoeller diagrams and autocorrelation length, this time calculated from the 100 m resolution simulations. We find larger areas covered by clouds and on the same time larger (drier) regions where no clouds from. The clouds of the local radiation simulations also contain more liquid water. The stronger development in the 3D *local* radiation case is evident.

### 3.2.4 Difference in the 100 m and 50 m resolution simulations and the performance of the NCA

The difference in the development in our simulation and the rather surprising result that 3D effects are stronger in the 100 m resolution simulation than in the 50 m resolution simulation can be related to the limitations of the Neighboring Columm Approximation.

The fact that the NCA uses only the direct neighboring column of a grid box to estimate the 3D cloud side cooling has two implications: First, the 'warming' effect of clouds nearby is neglected, which leads occasionally to slightly too high cooling rates. Second, the cloud side cooling is located at the outer most grid box of a cloud. As most of the cloud side cooling is located within the first 50 m of the cloud, this is still a reasonable assumption if the model resolution is not higher then 50 m. However, if clouds are thin in terms of optical thickness, the NCA misses some of the cooling which is in a real 3D thermal radiation simulation found further inside the cloud. As pointed out by Klinger and Mayer (2016), the uncertainty increases for horizontal grid box sizes of 50 m or smaller. By performing simulations at 50 m horizontal resolution, we push the NCA to its limits.

Looking into the data of our simulations of this study, we separated the number of cloud side grid boxes and cloud top grid boxes per total number of cloudy grid boxes at three time steps of our simulations (5 hours, 10 hours and 20 hours). As we increase horizontal resolution only, we expect a factor of four more cloudy grid boxes in the 50 m resolution simulations (assuming, that the total cloud volume remains the same). At the same time, the number of cloud side gird boxes increases only by a factor of two, the number of cloud top grid boxes increases, similarly to the total number, by a factor of four. It therefore follows that we find less cloud side grid boxes (per total number of cloudy grid boxes) in the 50 m resolution simulations. If the optical thickness of those cloud side grid boxes is small, we will therefore neglect some of the 3D cooling further inside the clouds in the 50 m resolution simulation.

To see if this is the case in our simulations, we extracted the number of cloud side grid boxes of an optical thickness < 1 in our cloud data. In the 50 m resolution simulations, about 30% of the cloud side grid boxes at 5 hours of simulation have an optical thickness lower than 1 while we find only 12 - 14% in the 100 m resolution simulation. In addition, the number of cloud side grid boxes at 50 m resolution is less (at 5 h about 5% in the 50 m resolution simulation and about 20% in the 100 m resolution simulation).

Therefore we neglect some of the 3D cooling further inside the clouds in the early part of our simulations. After clouds have grown and contain more liquid water, the NCA performs better in the 50 m resolution simulation as well. The exact data from our simulations are shown in Table 3.

We hypothesize therefore that the 3D cooling is, in an average sense, better calculated in the 100 m resolution simulations, 3D effects are stronger there or are earlier evident. In the 50 m resolution simulations, 3D effects are (due to the limitations of the NCA) weaker and closer to the 1D radiation simulations which explains the smaller differences between 1D and 3D thermal radiation at 50 m resolution.

## 4 Conclusions

We quantified the effect of thermal radiation on cloud development, comparing different radiation approximations. This was first investigated for idealized single clouds induced by a heat bubble perturbation close to the surface as well as for idealized simulations of a shallow cumulus cloud fields. Thermal radiation changes the cloud circulation by causing stronger updrafts and stronger subsiding shells around the clouds, which confirms previous results of Guan et al. (1997). However, we extended our study to a comparison of 1D and 3D thermal radiation and simulations of a shallow cumulus cloud field. Overall, we find increased mixing and entrainment in the simulations with radiation. Both the mixing and the resulting vertical velocities are due to a destabilization of the atmosphere which results from thermal cooling, as e.g. also found by Sommerai (1976), Fu et al. (1995), Petters et al. (2012) and Lilly (1988). Clouds also become deeper in vertical extent and contain more liquid water if thermal radiation is accounted for.

One important objective of our simulations was to investigate the effect of 3D *local* thermal radiation. Therefore, we performed five different thermal radiation simulations. We separate between 1D and 3D thermal radiation as well as between *local* and *averaged* radiation and a *constant cooling* simulation. We find that the effects described above are stronger if 3D thermal radiation is applied, compared to 1D thermal radiation. The most pronounced difference between the averaged and local radiation simulations and also between the 1D and 3D local radiation simulation is the different organization of the cloud field. In a local application of thermal radiation, clouds first organize in cell structures, similar to those generated with cold pool dynamics. However, as rain is switched off in our simulations, a different process is responsible for this cell development. To investigate how these cells really form will be studied in in a next step. In case of local 3D thermal radiation we find larger clouds at the end of our simulations. In the 100 m resolution simulations, even a single large cloud patch. The formation of cells is only found in the local radiation simulations, the larger clouds in the 3D local application. Simply adding more cooling to a simulation (as it is done in the averaged radiation simulations) does not at all reproduce the 3D results. The local additional cooling at the cloud side leads to the stronger subsiding shell and the different spatial distribution of the cloud field.

Obviously, the simulations shown in this study are in an idealized framework to omit feedback mechanisms which would occur otherwise. The fixed surface fluxes omit the surface flux feedback which was found e.g. by Muller and Held (2012) who proposed that this could be a reason for the organization of clouds. Another feedback mechanism which we neglect is the effect of rain and possible cold pool dynamics. Cold pool dynamics are usually associated with cloud organization (e.g. Seifert and

| Part of Cloud | Resolution | 50 m | | | 100 m | | |
|---|---|---|---|---|---|---|---|
| | Simulation Time | 5 h | 10 h | 20 h | 5 h | 10 h | 20 h |
| | Simulation Type | | | | | | |
| Cloud Top Fraction | NoRad | 28 | 22 | 18 | 29 | 19 | 16 |
| | 1D ICA | 31 | 18 | 15 | 30 | 15 | 13 |
| | 1D AVG | 31 | 18 | 16 | 29 | 16 | 14 |
| | 3D NCA | 30 | 18 | 16 | 29 | 16 | 14 |
| | 3D AVG | 31 | 18 | 14 | 30 | 15 | 13 |
| | const.cooling | 31 | 19 | 16 | 30 | 16 | 14 |
| Cloud Side Fraction | NoRad | 3 | 21 | 20 | 10 | 42 | 40 |
| | 1D ICA | 6 | 23 | 19 | 18 | 43 | 36 |
| | 1D AVG | 6 | 21 | 20 | 19 | 46 | 36 |
| | 3D NCA | 6 | 24 | 18 | 20 | 47 | 36 |
| | 3D AVG | 6 | 22 | 21 | 20 | 45 | 24 |
| | const.cooling | 5 | 24 | 23 | 16 | 46 | 44 |
| Cloud Side $\tau < 1$ | NoRad | 42 | 16 | 13 | 18 | 5 | 5 |
| | 1D ICA | 31 | 13 | 10 | 12 | 5 | 4 |
| | 1D AVG | 32 | 14 | 11 | 13 | 5 | 4 |
| | 3D NCA | 29 | 13 | 10 | 12 | 5 | 5 |
| | 3D AVG | 31 | 14 | 11 | 12 | 4 | 4 |
| | const.cooling | 33 | 14 | 11 | 14 | 6 | 4 |

**Table 3.** Percentage fraction of cloud side and cloud top grid boxes per total number of cloudy grid boxes at 5 h, 10 h and 20 h of the 50 m and 100 m simulations as well as the percentage fraction of cloud side grid boxes with an optical thickness (at 550 nm) lower than 1.

Heus (2013)). This cannot be the cause for the cloud organization which we find in our simulations. However, it is obvious that the clouds produced in our simulations are deep enough to cause rain from about 20 hours on, if we would allow it. If we would account for rain effects, the whole system would possibly change. Rain would set in earlier in all thermal radiation simulations (compared to the *No Radiation* case), and most likely earlier in the *local radiation* simulations. We rerun the *1D ICA local radiation* simulations, allowing for rain, and found rain to occur after 22 hours.

Previous studies (Muller and Held (2012), Emanuel et al. (2014), Wing and Emanuel (2014), Muller and Bony (2015)) used

RCE experiments and found thermal radiation to be a key driver for cloud organization. Our simulations are for a much smaller domain, with higher spatial resolution, and without deep convection. Yet, we also find that thermal radiation is a driver for organization. Also, our simulations show that it might be essential how radiative transfer is applied. The effects of *local* and *averaged* application of radiation differs significantly. While we find differences in the organization of the cloud field in the

5 *local 1D Thermal ICA* and *3D Thermal NCA* radiation simulations, the way how the *averaged* radiation is applied does not seem to yield significant differences. A similar result was also found by Cole et al. (2005).

Further studies are necessary to show the robustness of the results presented here and an improvement of the 3D radiative transfer calculations seems to be necessary for high resolution simulations. Additionally, future studies have to be extended to different cloud types, should also account for 3D solar radiative effects and different feedback mechanisms such as rain,

10 adjusting surface fluxes and more.

*Acknowledgements.* This work was funded by the Federal Ministry of Education and Research (BMBF) through the High Definition Clouds and Precipitation for Climate Prediction (HD(CP)$^2$) project (FKZ: 01LK1208A), HD(CP)$^2$ phase 2 (FKZ: 01LK1504D) and the DFG Transregio 165 "Waves to Weather". We thank George Craig for very helpful suggestions for the setup of the model experiments and the interpretation of the data. Many thanks go to DKRZ, Hamburg, for providing us with the computational resources. Last, we thanks three anonymous

15 reviewers, whose comments helped us improve our manuscript.

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

## List of Figures

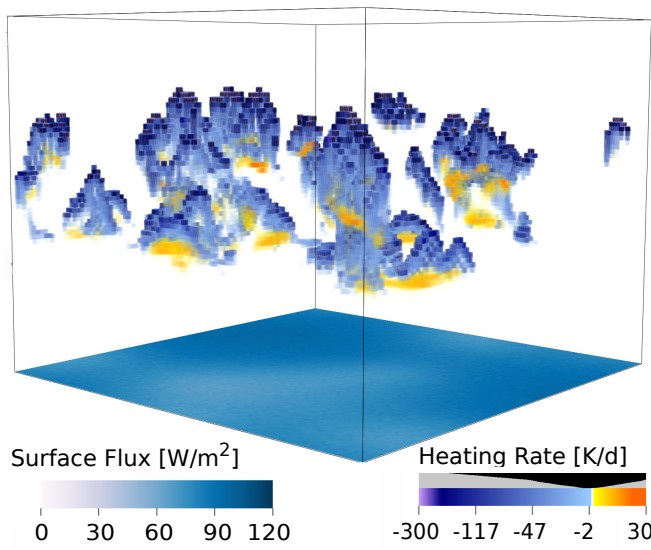

**Figure 1.** Three dimensional heating and cooling rates and surface fluxes in a cumulus cloud field, calculated with the Monte Carlo model MYSTIC (Mayer, 2009; Klinger and Mayer, 2014). The black and gray bar shows the opacity of heating rates.

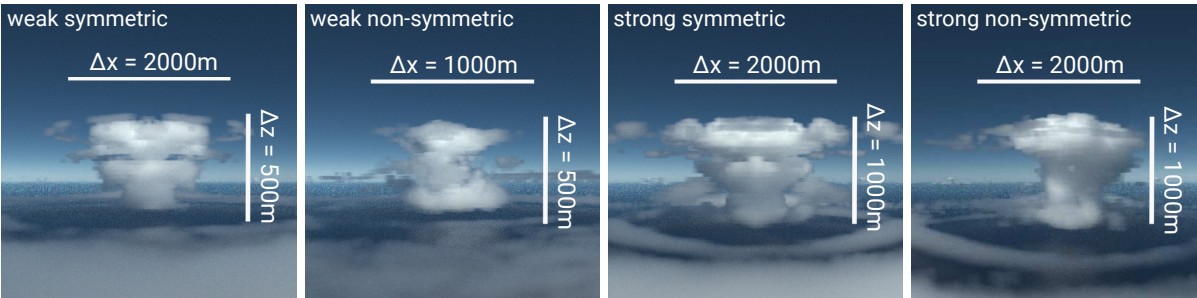

**Figure 2.** Visualization of the four single cloud simulations (first: weak symmetric; second: weak non-symmetric; third: strong symmetric; fourth:strong non-symmetric). The snap shot of the cloud field is taken at 40 min of the simulation of the *3D Thermal NCA* case. The visualization was performed with the 3D radiative transfer model MYSTIC (Mayer, 2009). The clouds in the background are a feature of the visualization and do not occur in the LES simulation of the single clouds. Clouds close to the surface are neglected in our analysis. Please note that the clouds are rather oblate, although they appear streched due to the perspective used in this visualization.

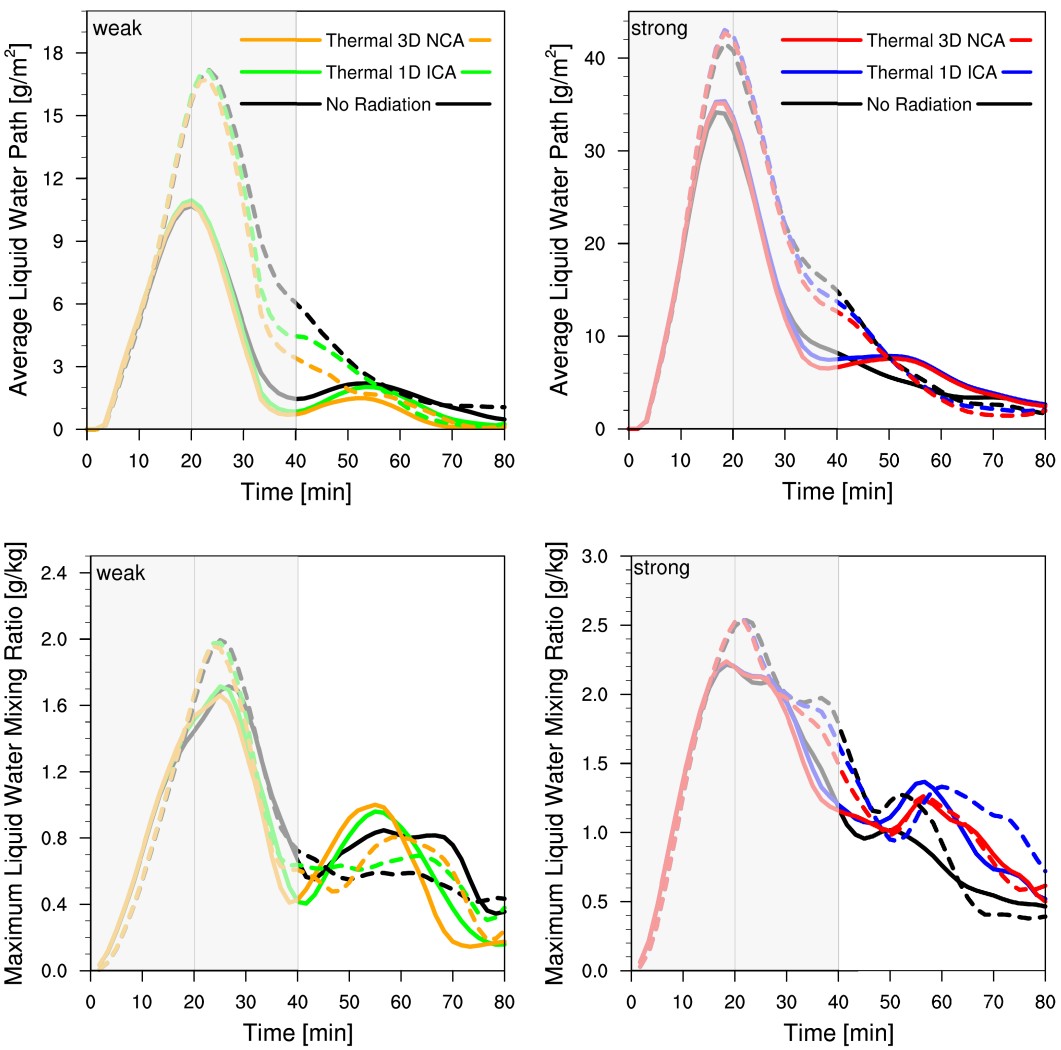

**Figure 3.** Time development of conditionally sampled liquid water path and maximum liquid water mixing ratio for the simulations of the single clouds. Only liquid water belonging to the single cloud was considered. The left column shows the weaker forced, the right column the stronger forced single clouds. Solid lines represent the non-symmetric cloud, dashed lines the symmetric cloud.

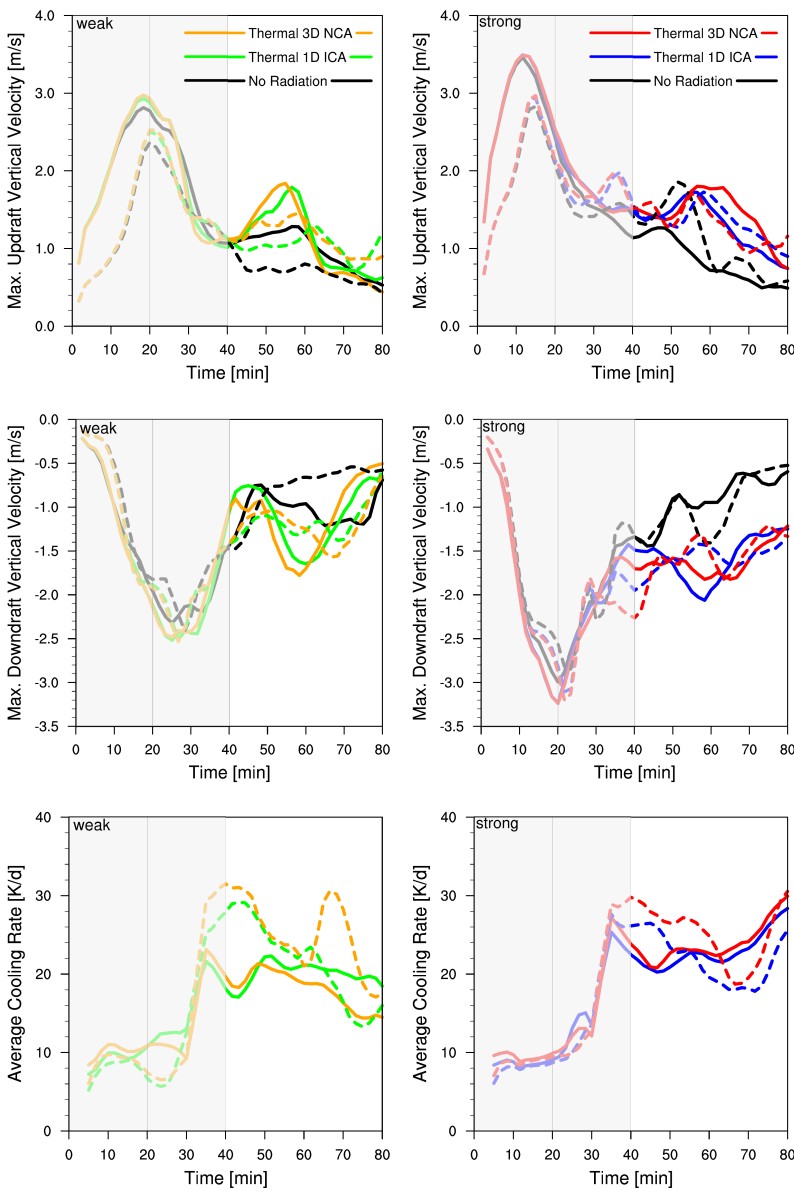

**Figure 4.** Time development of maximum updraft and downdraft vertical velocity as well as conditionally sampled cooling rates of the single cloud simulations for the simulations of the single clouds. Cooling rates were sample at the single cloud only. The left column shows the weaker forced, the right column the stronger forced single clouds. Solid lines represent the non-symmetric cloud, dashed lines the symmetric cloud.

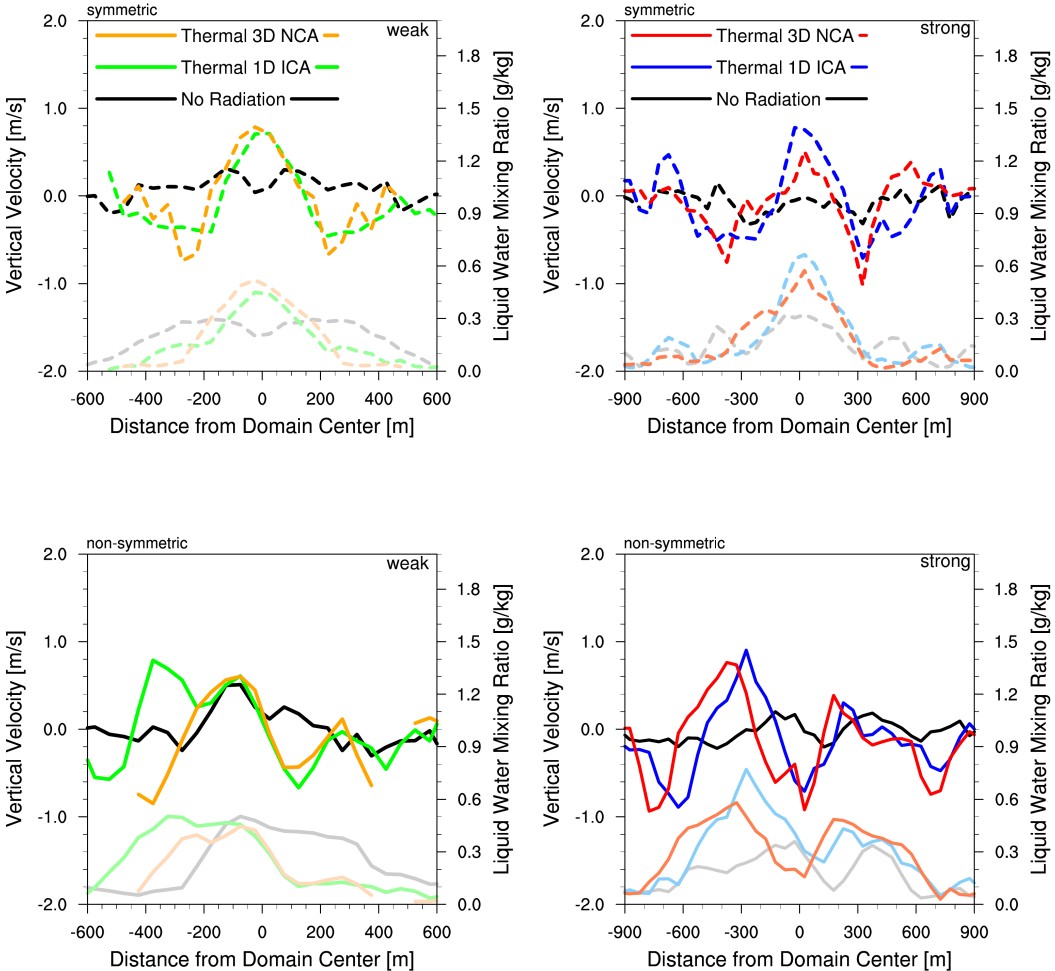

**Figure 5.** Height and time averaged transects of liquid water mixing ratio and vertical velocity. Liquid water mixing ratio is shown in pale colors on the bottom, the vertical velocity is shown in middle of each figure. The corresponding axes are on the right and left respectively. The time average was performed over 3 min at around 60 min simulations time. The vertical average was taken in the middle of the vertical extend of the cloud to cover cloud side areas. Dashed lines show the symmetric cloud, solid lines the non-symmetric cloud.

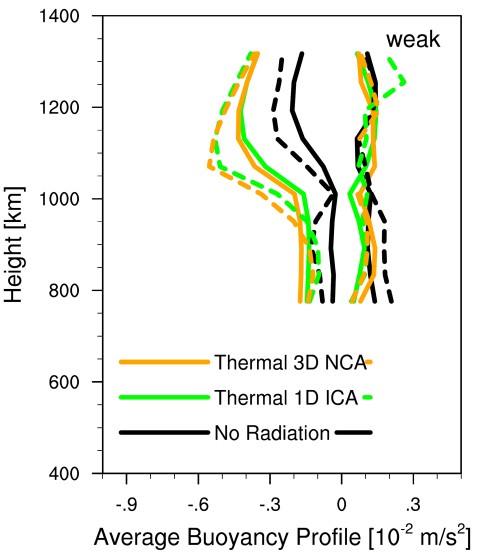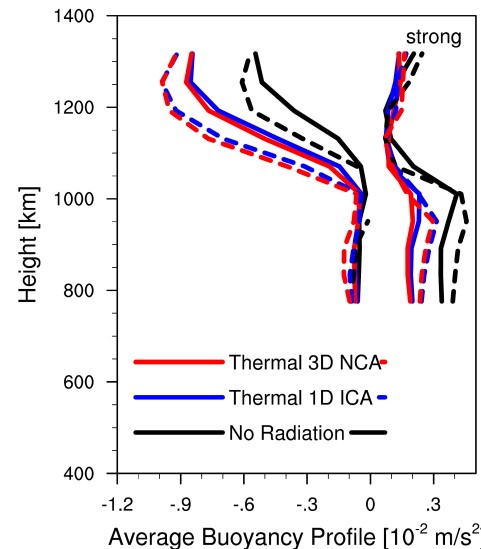

**Figure 6.** Positive and negative buoyancy profile sampled at 50-58 min of the simulation in the cloudy area. Dashed lines show the symmetric cloud, solid lines the non-symmetric cloud.

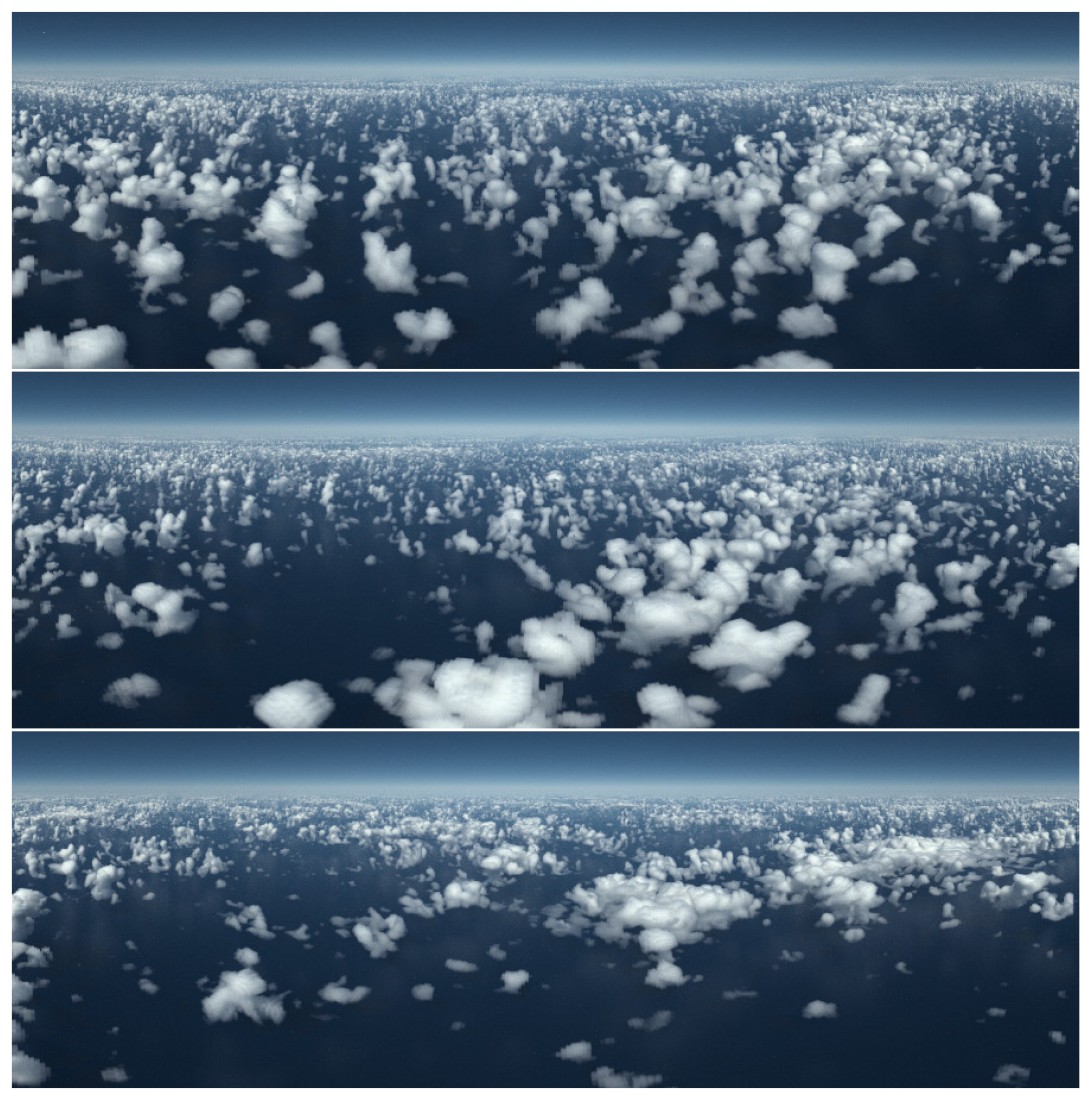

**Figure 7.** Visualization of the cumulus cloud field of the *constant cooling* (top), *3D Thermal Average* (middle) and *3D Thermal NCA* (bottom) simulations. The snap shot of the cloud field is taken at 20 hours of the simulation. The visualization was performed with the 3D radiative transfer model MYSTIC (Mayer, 2009).

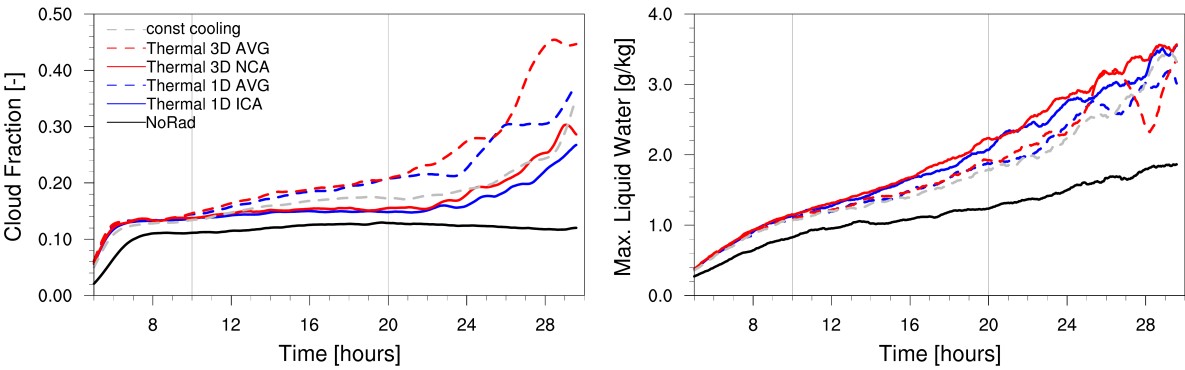

**Figure 8.** Time development of cloud fraction and maximum liquid water mixing ratio from 5 to 30 hours. The two gray lines (at 10 hours and 20 hours) separate the development in different periods for the further analysis.

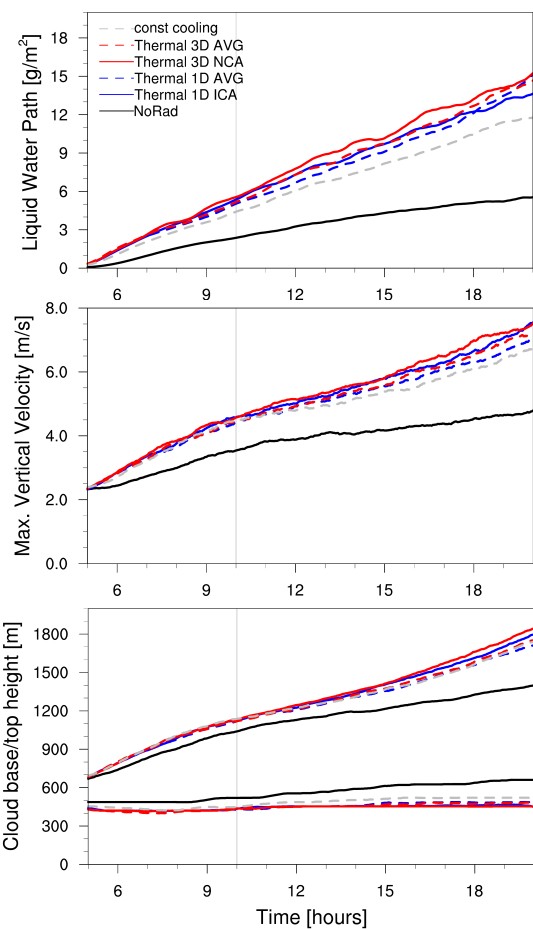

**Figure 9.** Time development of liquid water path, maximum vertical velocity and cloud base and cloud top height. The gray line (at 10 hours) separates the development in different periods for the further analysis.

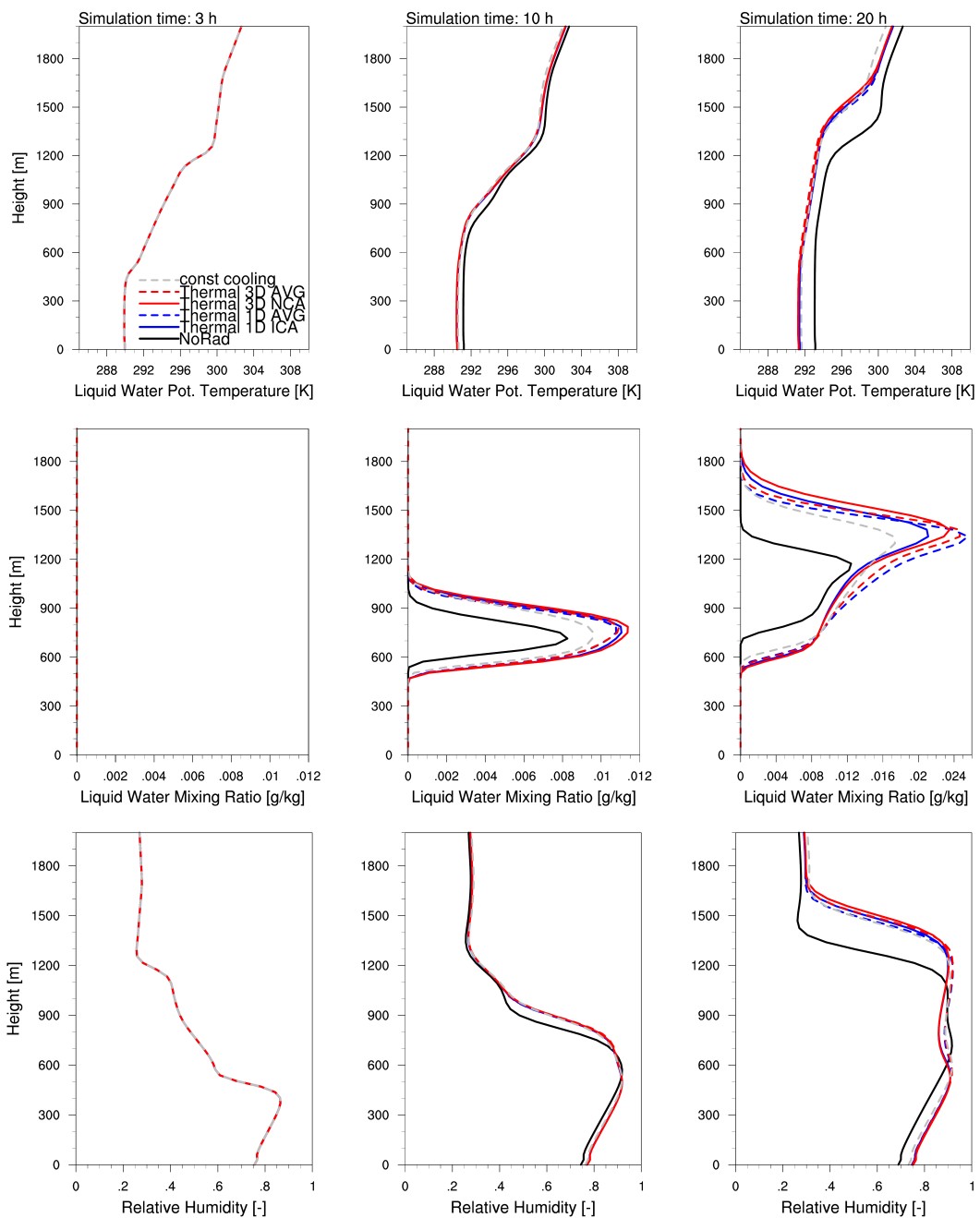

**Figure 10.** Time averaged profiles of liquid water potential temperature, liquid water mixing ratio and relative humidity. The profiles are shown at the restart time (3 hours), as an 3 hour average centered at 10 h and 20 h of the simulations.

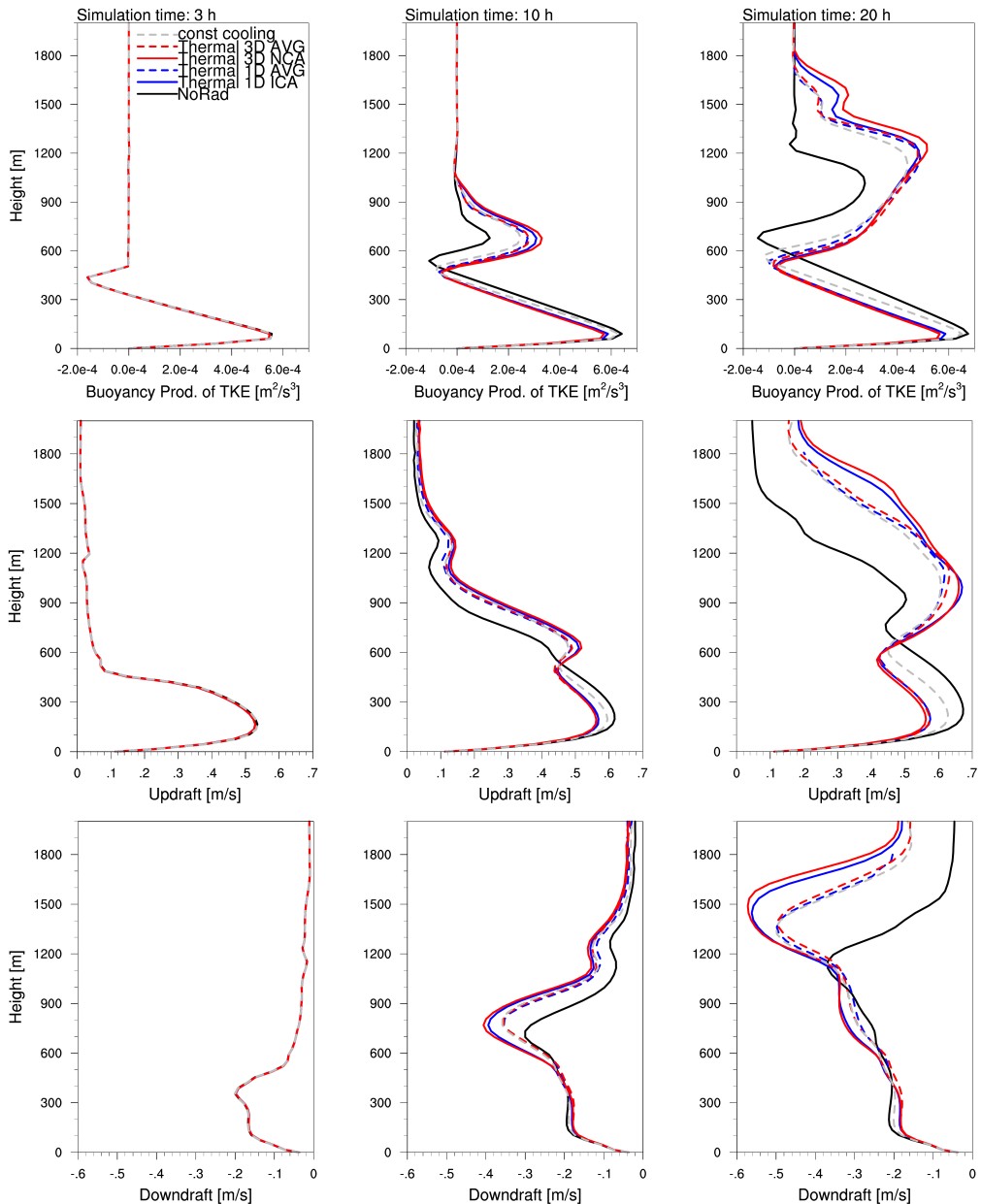

**Figure 11.** Time averaged profiles of buoyancy production of the TKE and updraft and downdraft vertical velocity.The profiles are shown at the restart time (3 hours), as an 3 hour average centered at 10 h and 20 h of the simulations. Updraft and downdraft vertical velocities were extracted from the 3D data following Park et al. (2016)

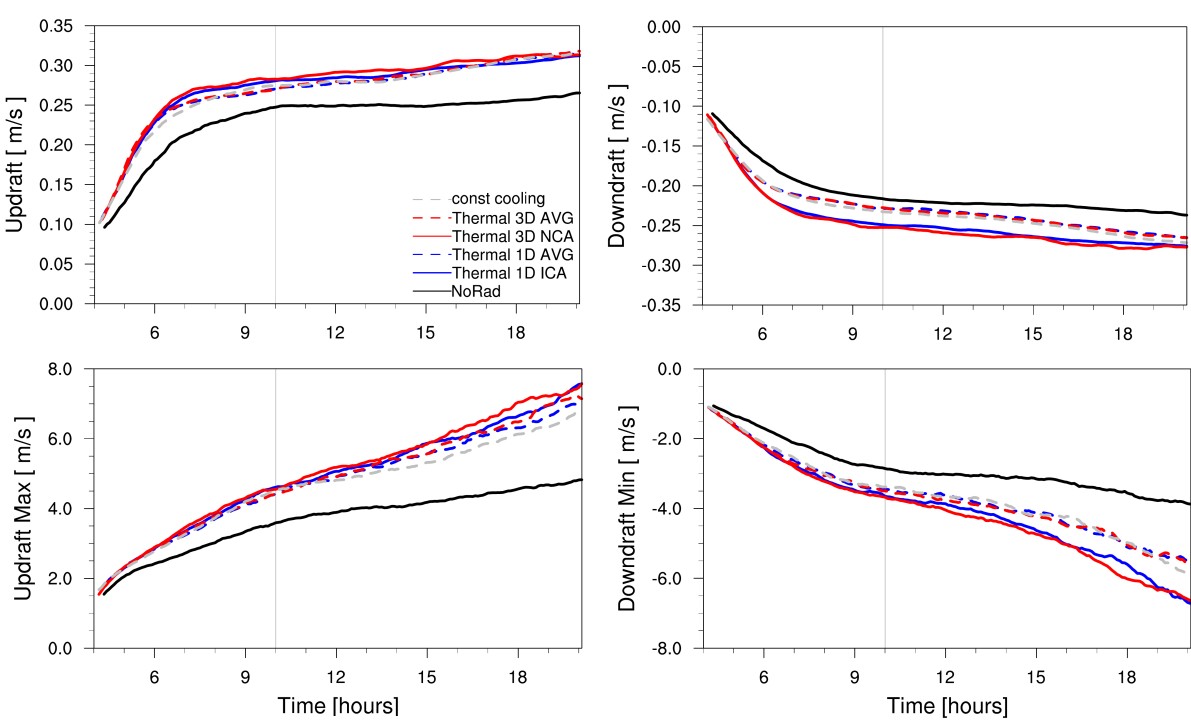

**Figure 12.** Time series of averaged vertical velocity as well as maximum updraft and downdraft vertical velocity of updrafts and downdrafts.

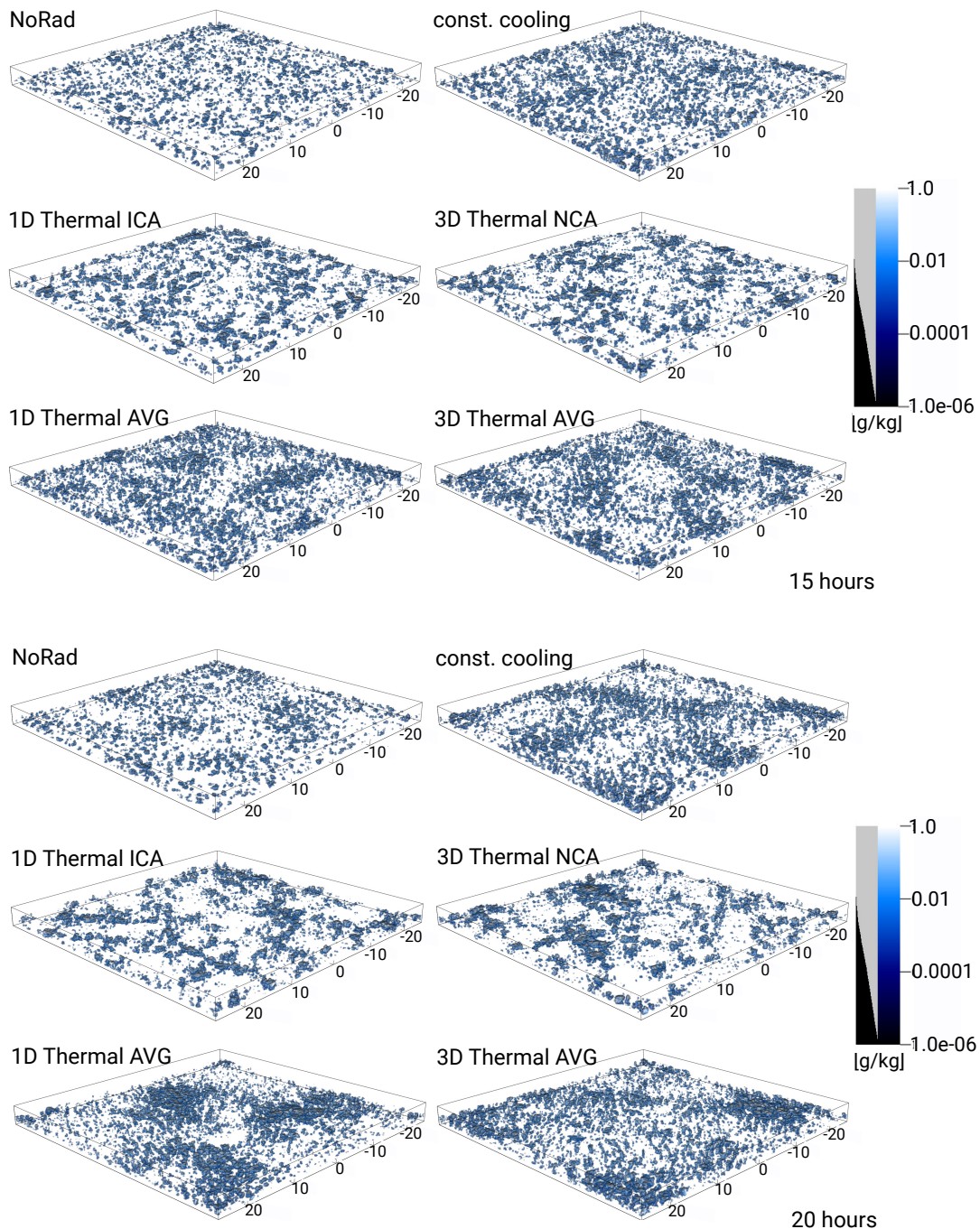

**Figure 13.** Cloud fields at 15 h (top) and 20 h (bottom) of the 50 m resolution simulation. The quantity shown is liquid water mixing ratio. The black and gray bar shows the opacity used in this visualization.

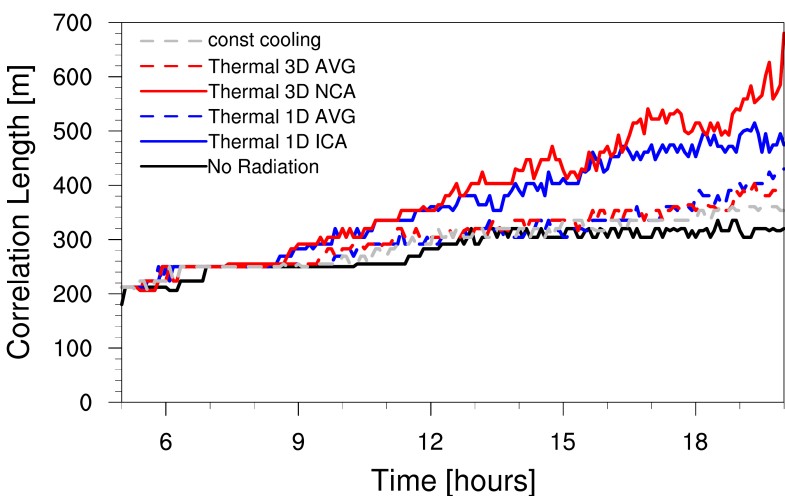

**Figure 14.** Time development of correlation length. The correlation length is defined by the shift where the correlation coefficient drops below 1/e.

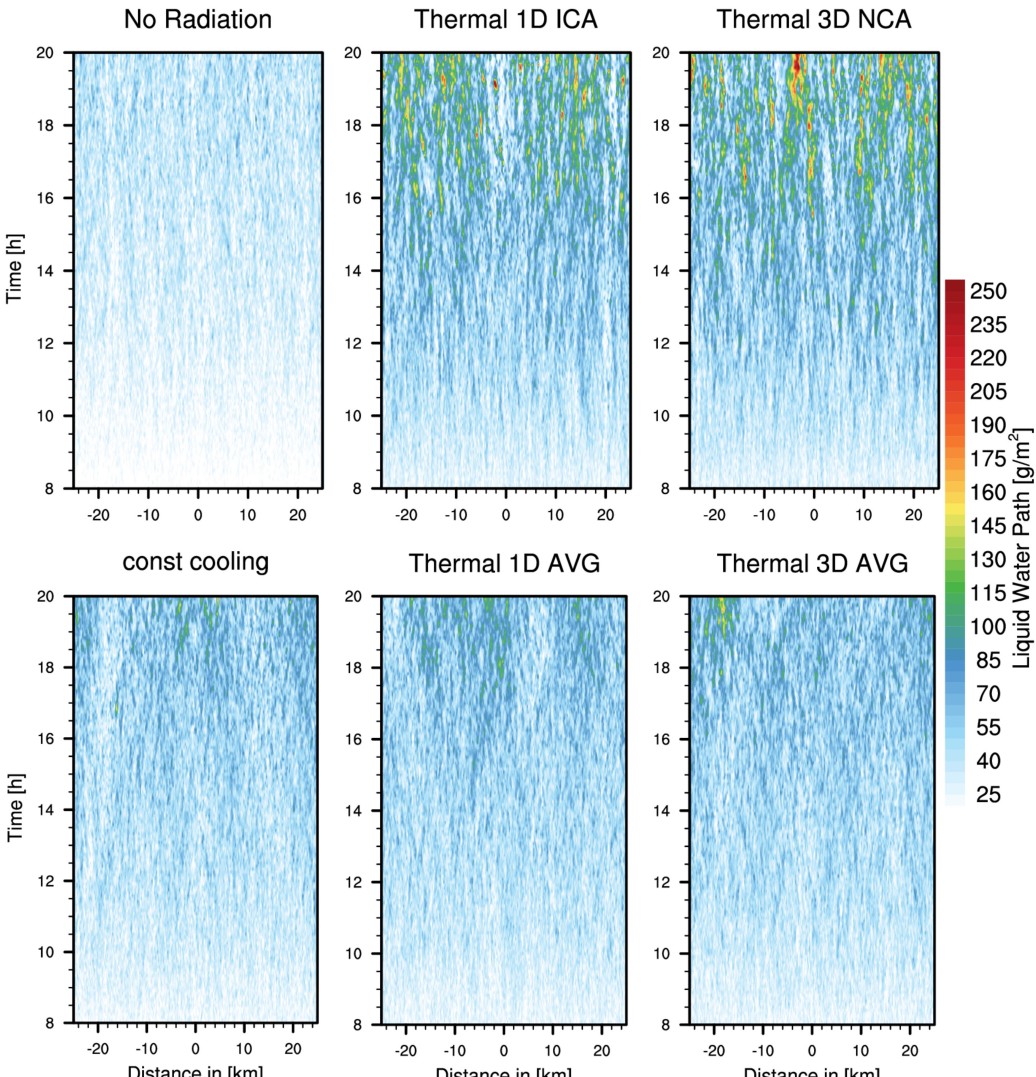

**Figure 15.** Hovmoeller Diagram of liquid water path, averaged in x-direction.

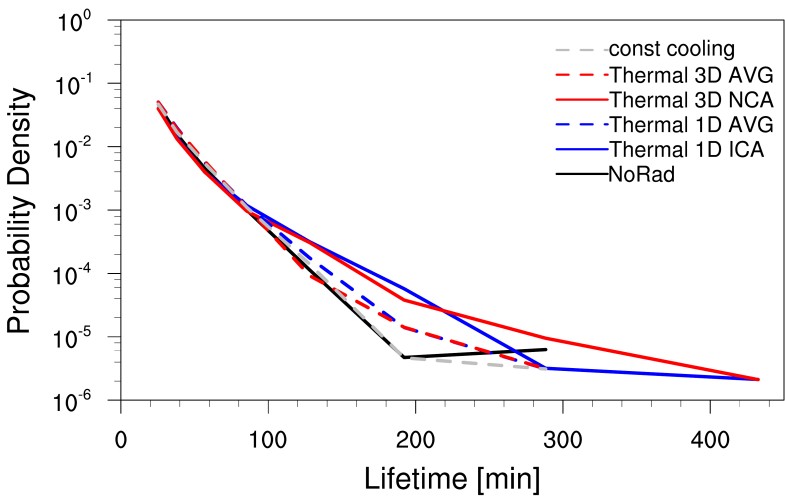

**Figure 16.** Probability density function of cloud lifetime: The lifetime of each cloud detected by the tracking algorithm was calculated within the first 20 hours of the simulations.

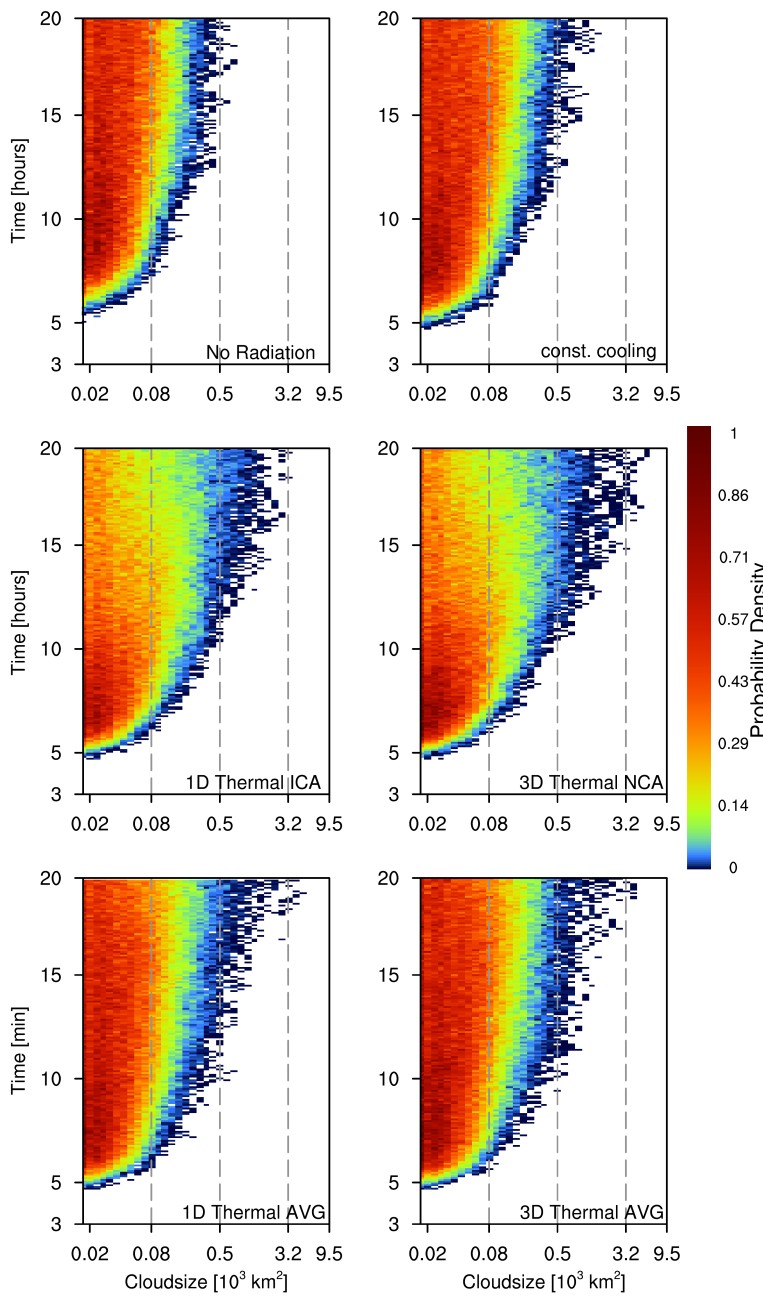

**Figure 17.** Probability density with respect to cloud size and the time of occurrence in the different simulations.

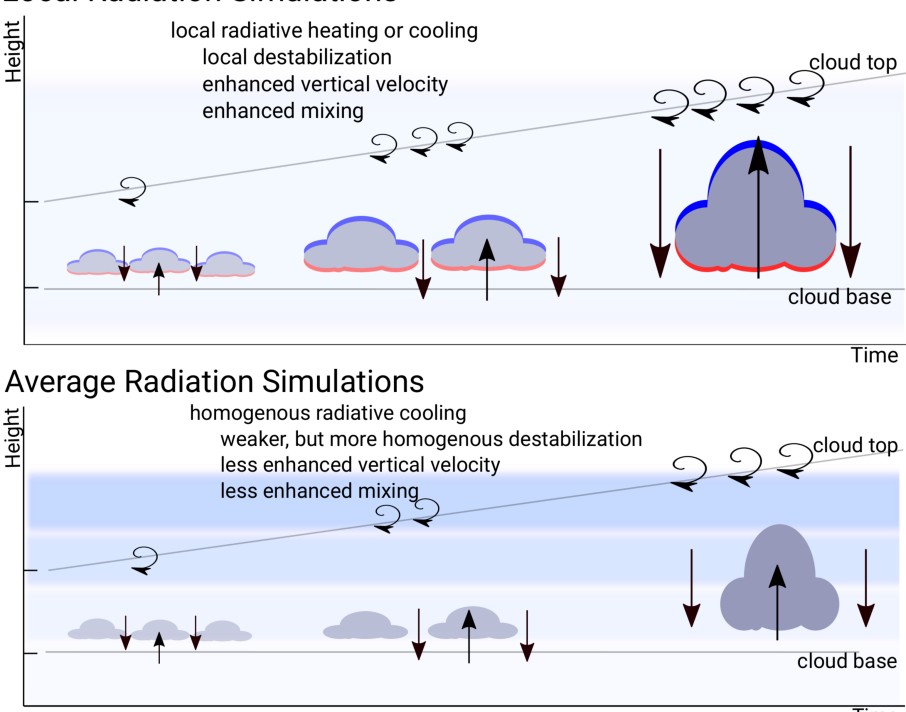

**Figure 18.** Schematic figure of the effects of thermal radiation in the presented simulations. The figure summarizes cloud development over time and height, showing the enhanced cloud growth, the development of vertical velocity (arrows), a deepening of the cloud layer and the enhanced mixing. Blue and red colors show thermal heating and cooling, either at the clouds itself (for *local radiation*) or *averaged* in the cloud layer.

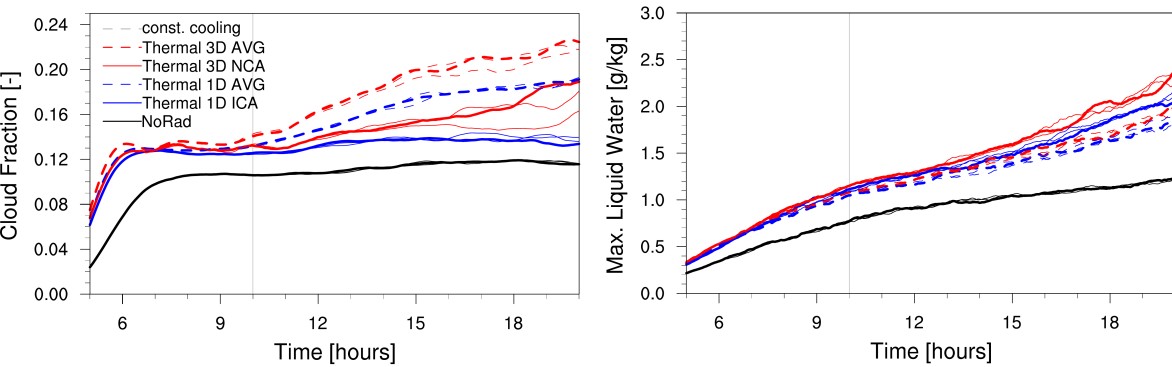

**Figure 19.** Time development of cloud fraction and maximum liquid water mixing ratio from 5 to 20 hours. Thin lines show the results of two additional simulation respectively.

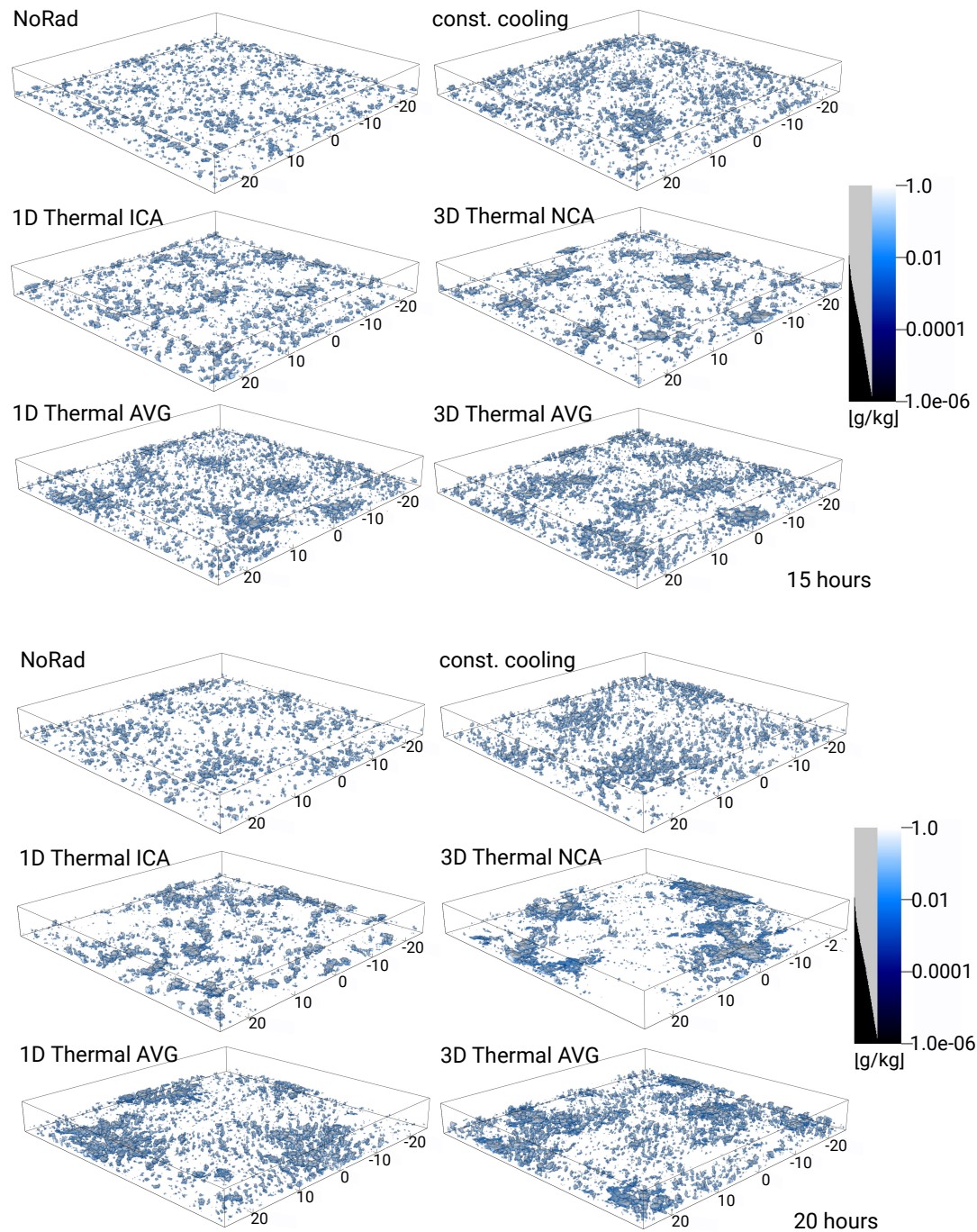

**Figure 20.** Cloud fields at 15 h (top) and 20 h (bottom) of the 100 m resolution simulation. The quantity shown is liquid water mixing ratio. The black and gray bar shows the opacity used in this visualization.

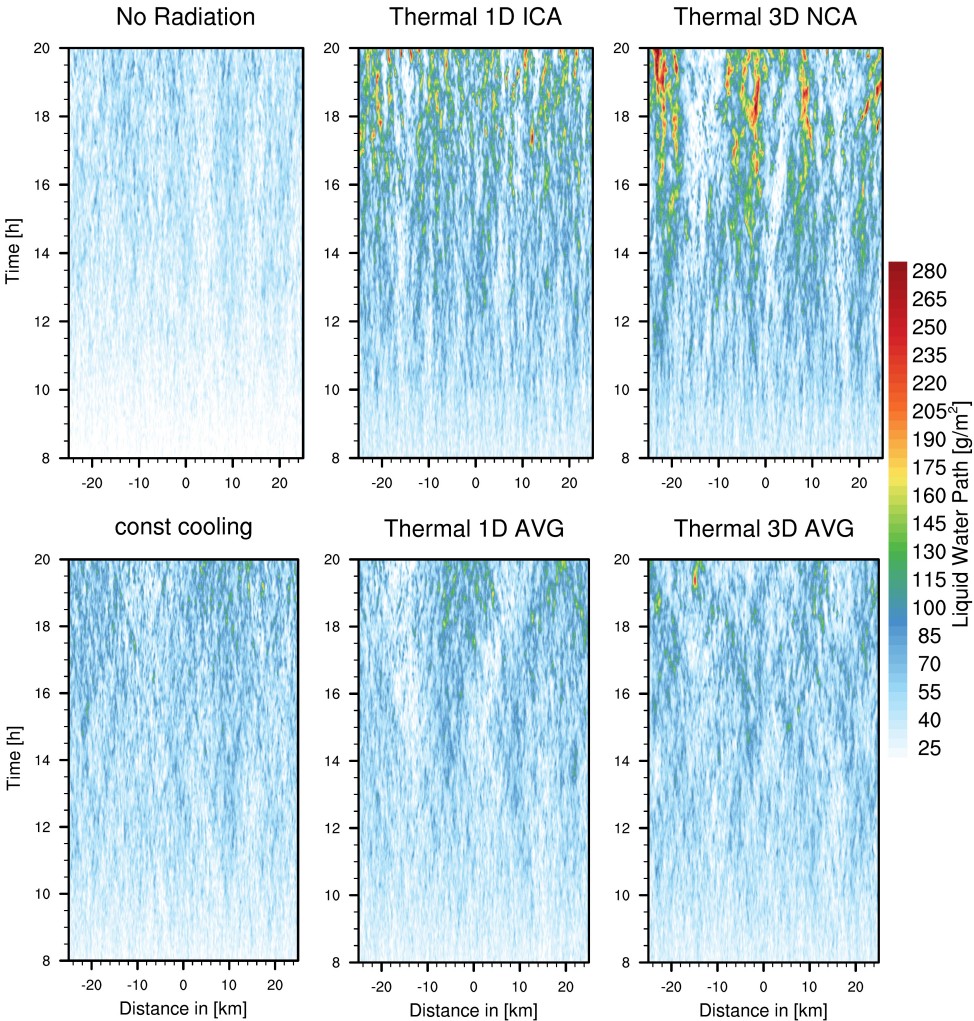

**Figure 21.** Cloud organization in the 100 m resolution simulations shown as Hovmoeller diagrams of the liquid water path, averaged in x-direction

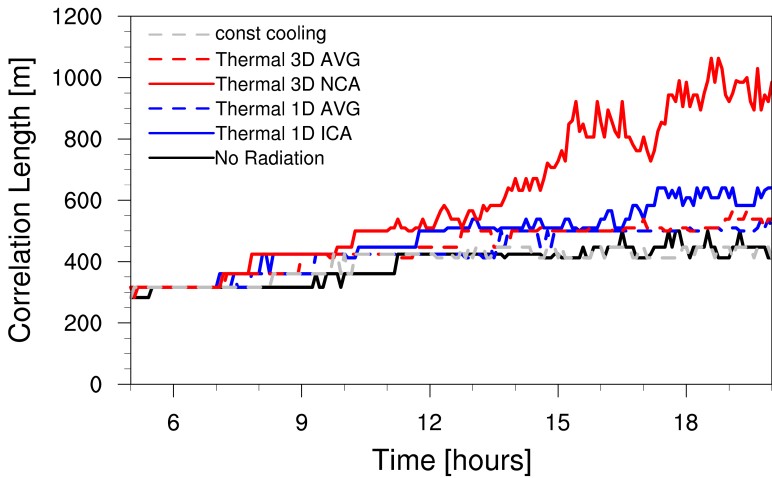

**Figure 22.** Temporal development of the correlation length for the 100 m resolution simulation. The correlation length is defined by the shift where the correlation coefficient drops below 1/e.