# Peer review of "Effects of 3D Thermal Radiation on the Development of a Shallow Cumulus Cloud Field"

_Atmospheric Chemistry and Physics, 2016_

## Referee Comment (RC1) · Anonymous Referee #3 · 7 Dec 2016

Summary:

The stated goal of this paper is to illustrate the effect of 3D thermal radiative transfer on simulated clouds. This is done by simulating a single plume cloud and a field of oceanic cumulus clouds using suite of thermal radiation configurations, ranging from no radiation to 3D thermal radiation. While this is an interesting study it is not clear what new information is brought forth. There are papers in the literature which discuss the effect of turning radiation on and off and the effect of using average instead of local radiation. New results should be the effect of interactive 3D thermal radiative transfer on the simulation but the author has missed, or at least not referenced, a key paper which has presented this sort of experiment. In addition, the results from single simulations are not particularly convincing of a clearly different response when using 3D, versus ICA, thermal radiative transfer.

[Figure]

With these comments in mind, I suggest that the authors perform a major revision of the paper taking into account the comments below and previous results in the literature.

General comments:

Single cloud results: I would say that the results when using 1D and 3D thermal RT are almost indistinguishable and it would be challenging to read much into these results. For example, if you performed the simulations with 1D thermal RT again, or several times, but with a small random perturbation, I suspect the results would be about as different as that between the 1D and 3D. The main thing I can take from these simulations is that interactive "local" radiation has an influence versus no radiation but the complexity of the radiation parametrization doesn't seem to be too important, at least for the variables included in the analysis.

Cloud field results: I have the same comments here as for the single cloud experiment. For most of the variables the differences between results for 1D and 3D "local" thermal radiation are very similar. Is there an expectation that ensembles of simulations would show the differences to be statistically significant?

Specific comments:

Title: It does not accurately reflect of the contents of the paper. The 3D thermal radiation is a relatively small part of the paper and the paper focuses on a very particular types of cloud. I.e, I don't think the results could be generalized to all clouds.

References: The highly relevant paper by Mechem is missing:

Mechem, D. B.; Kogan, Y. L.; Ovtchinnikov, M.; Davis, A. B.; Evans, K. F. & Ellingson, R. G. Multidimensional Longwave Forcing of Boundary Layer Cloud Systems Journal of the Atmospheric Sciences, 2008, 65, 3963-3977.

There are some papers that have discussed interactive cloud resolving simulations considering aspects of 3D solar radiative transfer:

[Figure]

Koracin, D.; Isakov, V. & Mendez-Nunez, L. A cloud-resolving model with the radiation scheme based on the Monte Carlo method Atmospheric Research, 1998, 47-48, 437-459

Frame, J. & Markowski, P. Numerical Simulations of Radiative Cooling beneath the Anvils of Supercell Thunderstorms Monthly Weather Review, 2010, 138, 3024-3047

And there are papers that discuss the effect of using domain mean radiative fluxes (here I give just two examples, I am sure there are others),

Petch, J. C. and Gray, M. E. B. (2001), Sensitivity studies using a cloud-resolving model simulation of the tropical west Pacific. Q.J.R. Meteorol. Soc., 127: 2287–2306. doi:10.1002/qj.49712757705

Cole, J. N. S.; Barker, H. W.; Randall, D. A.; Khairoutdinov, M. F. & Clothiaux, E. E. Global consequences of interactions between clouds and radiation at scales unresolved by global climate models Geophysical Research Letters, 2005, 32, L06703

Introduction, long paragraph stating at line 3, page 2: This paragraph is challenging to read and needs to rewritten since in its current state it comes across as an "information dump". From it reader needs to pull together information needed for the remainder of the paper. Breaking the paragraph into at least two would help as would putting the information into an order that fits with rest of the paper. I.e., general effect of thermal radiation on cloud development, previous results 1D versus 3D thermal radiation and a justification for examining local versus non-local radiation (1D versus slab averages) with discussion of previous results.

Page 2 line 14: The discussion of the Guan study is a bit unclear since that study compared 3D thermal radiation against the case of no radiation, not versus 1D radiative transfer,

Page 3, paragraph at line 5: Did you add modify the equations used for the cloud microphysics to explicitly model enhanced emission by drops? My understanding of

the papers by Harrington is that a term for the radiative heating and cooling of the drop is considered. If you did not add drop cooling to the microphysics the interpretation of this paragraph is tricky.

Page 4, line 23: Is the shape of the cloud that sensitive to the structure of the perturbation?

Page 5, line 27: Why not show the clouds simulated using the 3D thermal RT? Would it not be the most realistic? Also showing the cloud field at the 20 minute point does not make sense given the discussion in the text. The text it is pointed out that it is cloud field in the period 40-80 minutes that are to be the focus of the analysis. Why not show the cloud at that point?

Page 6, line 13: The meaning of this statement 'from about 30 min onward the cloud stays rather constant at a certain height' is not clear. What exactly stays constant (liquid water path?)?

Page 6, line 15: This sentence,

"All simulations show that the liquid water path (top row of Fig. 3) is reduced by thermal radiation in this "second stage" (from about 20 min to 40 min)."

is not clear. The same reduction is seen in all simulations without any radiation. Do you mean to say that thermal radiation causes liquid water path to be less than the case with no radiation? This difference is pretty small.

Page 7, lines 1-27: This analysis seems to end abruptly or I'm missing something. There is an idea of "subsiding shells" to explain the subsidence around the edge of the cloud. As mentioned in the text, Heus and Jonker, 2005 attribute the presence of the shell to "negative buoyancy, resulting from evaporative cooling following lateral mixing of environmental air with cloudy air.". The results with thermal radiation have downdraft shells that are stronger than that in the no-radiation case.

Is it not possible to use the output from the model to further analyze and show why

the shell is enhanced in the presence of thermal radiation? Is it the radiation directly producing more negative buoyancy, Figure 1 suggests large radiative cooling, or does it induce an environment that enhances the evaporative cooling? It must be possible to quantify statements like "This might be due to the thermal cooling at cloud tops, and in case of 3D Thermal NCA radiation at cloud sides." and "The stronger horizontal buoyancy gradient (difference between positive and negative buoyancy in Fig 6) generates enhanced turbulence and therefore stronger evaporation.".

Section 3.2: What is special about the "restart time" at 3 hours? Was the model run differently up to this point?

Page 8, line 15: Are the liquid water path and other variables shown for this case averaged over the entire domain or sampled only over clouds?

Page 8, line 17: How robust are these results?

Page 8, line 19: Is it an expected result that "All quantities increase over time.". If you continued running the simulation would it go into a quasi-equilibrium state?

Page 8, line 20: Remove this sentence as it is obvious,

"The different development of the No-Radiation simulation and the radiation simulations is related to the missing cooling of the thermal radiation in the No-Radiation simulation."

Page 8, line 23: Why does the lack of thermal radiative cooling lead to a higher cloud base? It is not clear to me.

Page 8, line 25: I don't think you want to use the term "bias" here, perhaps the word "change" instead?

Page 8, line 26; Perhaps a clear term than "interactive" would be "local" since the "averaged" radiation is also interactive since it still reacts to changes in the clouds.

Page 8, line 29: What is so interesting about the liquid water path and maximum liquid water content?

Page 9, line 4: The rate of increase in cloud fraction for the "interactive" radiation after hour 22 is nearly as large as for the averaged radiation. How does this fit into the organization hypothesis?

Page 9, line 13: Initial profiles in first column not first row.

Page 9, line 26: How significant is the approximately 5% greater liquid water content in the 3D NCA simulation at hour 10?

Page 9, line 33: Figure 11 first column, not Figure 11 (second column)?

Page 10, line 1: The 3D NCA simulations produce slightly more TKE through buoyancy in the upper cloud with stronger upward and downward vertical winds. Again, is this significant? As discussed further down in this section the more significant result is that horizontal averaging causes more significant differences.

Page 10, line 24: From the results shown I would suggest that it is not conclusive that "3D Thermal NCA" increases the results shown. The differences relative to 1D ICA are quite modest and it is not clear if they are by chance or systematic.

Page 11, line 2: spacial -> spatial

Page 12, line 14: Can you quantify that "we find larger structures earlier in the 3D Thermal NCA radiation simulation"? Staring at the plots for 1D ICA and 3D NCA in Figure 14, it is not clear how one objectively comes to this conclusion. The color contouring gives some bias toward clouds with larger liquid water path, not necessarily larger cloud structures.

Section 3.2.3: Were 3 simulations performed for each radiation configuration? If so, did the "small differences" lead to a stronger or weaker case for differences between the 1D and 3D interactive simulations? The results of the simulations, especially 3D NCA, seems rather sensitive to horizontal resolution (Figs. 17 and 18). Any speculation as to why? Should we expect different results if we reduced the horizontal resolution to 25 m?

Page 14, line 25: It is not a strong or clear result in this paper that the 3D interactive radiation is significantly stronger than the 1D ICA radiative transfer. Therefore, this statement is not well supported.

Table 1: Horizontal resolution does not match with text, 100 m in table and 50 m in text. If it is the latter then number of gridboxes is incorrect since domain size is quoted in text to 6.4 km by 6.4 km.

Figure 5: It is very difficult to read this figure. For example, the dashed lines are almost impossible to see on the printed document. For this figure titles indicating which are symmetric cloud and which are non-symmetric clouds is warranted.

Figures 10 and 11: The solid versus dash in the legend is very difficult to make out.

---

## Referee Comment (RC2) · Anonymous Referee #1 · 14 Dec 2016

Summary This is a potentially interesting paper that may be publishable after suitable major revision. It would benefit from a better focus on its main points that would clarify the contribution of the paper to original knowledge. The simple finding that 3D is different from 1D, and that 3D is needed to improve the development of model clouds is not, in itself, an original finding. Neither are most of the findings regarding changes in cloud circulation, liquid water content and lifetime, but these findings may be helpful in the context of confirming the earlier work of others. The finding that thermal radiation, when correctly treated in a 3D framework, triggers the organization of clouds appears to be the main result, and if so, should be presented as such, but with greater clarity about what is meant by 'organization'.

[Figure]

Major revisions 1. The Introduction is not helpful in its present form. There is a jumbled litany of past work that should be more critically presented: a lot refers to standard 1D theory that carries over to cloud development in general. This should be presented separately from the past findings on 3D thermal influences, both on individual cloud development and on cloud fields. Despite the repeated assertion that studies accounting for 3D effects are rare, insufficient acknowledgement is given to the earlier work. The earliest 3D calculation of the cooling rates from the sides of an isolated cloud was probably that of Harshvardhan et al. [JAS, 1981]. Mechem et al. [JAS, 2008] showed the importance of multidimensional radiative transfer to the forcing of large cloud systems.

The Introduction should acknowledge past work and provide motivation for why another such study is warranted. The goal seems to be limited to one brief sentence [p3, l12] that lacks specifics.

2. The Conclusion is not helpful in its present form. It should not be a simple summary of what was done, but rather should conclude what the results contribute to the advancement of knowledge. Have these results simply confirmed what others have previously found [p13, l15-l19]? Have these uncovered something new [organization?].

3. The discussion about resolution and reproducibility [p13, l1-l12] raises additional questions that should be addressed. If the effects are more pronounced at coarser resolution, would they be even smaller if a higher resolution than 50 m had been used? Davies and Alves [JGR 1989], for example, showed that 100 m is far too coarse to capture the peak cooling rates, and even 50 m underestimates the peak rate. This may not matter as much for cloud-tops, but will have a big effect on cloud-side cooling, which is the main novelty of 3D thermal calculations. Since 3D thermal effects should be larger at higher resolution, which is the opposite of the model results presented here, there is something counter-intuitive going on that should be addressed in the text. Are the results perhaps sensitive to initial conditions, requiring multiple samples to reach a firm conclusion? I find this a little troublesome.

---

## Referee Comment (RC3) · Anonymous Referee #2 · 6 Jan 2017

This manuscript addresses an important yet poorly understood topic by examining the impact of 3D longwave radiative processes on cloud development for small cumulus clouds. The methodology is appropriate and I believe the paper will make a valuable contribution to the community, but the presentation needs significant improvement before publication. Please find below a list of specific comments. In compiling the list I tried to avoid repeating earlier comments made in the interactive discussion, but some inadvertent repetitions may occur.

General comments:

The paper should comment on whether the results are likely to be affected in a significant way by any inaccuracies in its 3D radiation scheme, the Neighboring Column Approximation.

The summary section should mention that examining additional LES scenes is a key topic for follow-up studies (alongside with incorporating solar radiation, etc.), as the representativeness of current results can be established only by examining further scenes.

Specific comments:

Page 1, Lines 9-10: The meaning of "slab-averaged applications" is not clear to me. Also, the comma after "profile" in Line 11 is not needed.

Page 1, Lines 16-17: It would be important to clarify right at the first mention what is meant by "organization" and/or "organization effects" (e.g., fewer but larger clouds).

Page 5, Lines 6-7: It would help to point out that averaging is over the entire scenes, including even cloud-free grid cells. (This is clarified in the last sentence of Page 13, but readers may wonder well before that.)

Page 5, line 11: I suggest deleting the sentence "The overall cooling in a modeling domain is generally stronger in case of 3D thermal NCA radiation", as the results are discussed and explained later, while this section discusses only the experimental setup.

Page 6, Line 6: I wonder in what sense does cooling compensate for the temperature perturbation.

Figure 3 and most subsequent figures: Using longer dashes in all figures would really help, as I could distinguish dashed lines from solid ones only after strong zooming.

Figure 5 caption: It is not quite clear to me what "bottom" and "middle" mean in "bottom right axis" and "middle left axis."

Page 10, line 6: It might be worth pointing out that relative humidity is lower in interactive simulations even though the temperature is also lower, because the liquid water mixing ratio is higher.

Figure 10-11 captions: For clarity, I suggest replacing "as an 3 hour averaged" by "and as 3 hour averages centered" (perhaps using "starting" instead of "centered").

Page 12, Lines 27-29: It seems worth mentioning that the issues of cloud organization and the size-dependence of cloud lifetime are closely related, as smaller clouds dying and larger clouds growing will result in fewer but larger clouds and in longer correlation lengths. Also, it seems worth pointing out explicitly that it is the same entrainment-invigoration due to 3D interactive radiation that reduces cloud diameter for the cylindrical cloud and erodes small-size clouds for the LES cumulus scene (if this is correct).

Page 13, Lines 7-8: I don't quite understand the sentence "The separation into moist and dry regions is stronger in the simulation with a coarser resolution.", and so clarification would be helpful.

Page 13, Line 18: The comma after "both" can be deleted.

Page 13, Lines 27-29: It seems more important to emphasize the behavior before (rather than after) 20 hours, as that is the time period for which cloud organization results are presented (Figures 13 & 14). The time after 20 hours may be mentioned in passing, but the key point is that in the first 20 hours, clouds are larger in the interactive runs.

Page 14, Lines 17-18: The sentence "...it is not certain that we would ever reach the stage where clouds organize in the averaged radiation simulations, but we may reach the stage in the interactive ones" is confusing, because the paper discussed cloud organization in Section 3.2.2 and did not find it negligible in the interactive simulations. Page 14, Lines 3-5 also talk about significant cloud organization.

Page 14, last sentence of summary: This is a very important sentence, and even I suggest directly pointing out its main implication, that the impact of 3D effects comes from changing the spatial distribution (and not the mean value) of cooling.

Figure 19 is very helpful and I would even consider bringing it earlier, accompanied by some discussion of the key processes involved. For example, it could help to point out that the difference between 3D averaged and 3D interactive simulations is determined by the balance of two competing processes. In interactive runs, the stronger entrainment caused by cloud side cooling shrinks clouds, while the lack of cooling in the middle of updraft pockets leads to stronger updrafts and helps clouds grow. The balance of these two processes varies with the perimeter to area ratio of updrafts, and so the first process can be expected to win for small clouds, and the second one for large clouds. Finally, a minor point is that it would help to include a title for each panel or to specify in the caption what the top and bottom panels are for.

Appendix: I don't think there is a need for a separate Appendix, as the current Appendix contains only the two tables that could easily be moved into the main body. Also, it would be important to clarify what is meant by "vertical stretching".

---

## Author Comment (AC1) · 18 Feb 2017

We thank reviewer 1 for his helpful comments. The response is uploaded as a supplement, along with the revised manuscript and a difference-pdf.

Please also note the supplement to this comment:
http://www.atmos-chem-phys-discuss.net/acp-2016-896/acp-2016-896-AC1-supplement.zip

---

## Author Comment (AC2) · 18 Feb 2017

**Response to the comments of reviewer 2**

*Our responses are marked in italic and color.*

*We thank reviewer 2 for his helpful comments which helped us to improve the manuscript.*

This manuscript addresses an important yet poorly understood topic by examining the impact of 3D longwave radiative processes on cloud development for small cumulus clouds. The methodology is appropriate and I believe the paper will make a valuable contribution to the community, but the presentation needs significant improvement before publication. Please find below a list of specific comments. In compiling the list I tried to avoid repeating earlier comments made in the interactive discussion, but some inadvertent repetitions may occur.

General comments:

The paper should comment on whether the results are likely to be affected in a significant way by any inaccuracies in its 3D radiation scheme, the Neighboring Column Approximation.
* * *
*We added an additional subsection, explaining the general performance of the NCA, as described in Klinger and Mayer, 2016 (JQSRT). We also addressed the performance of the NCA in the context of this paper.*

The summary section should mention that examining additional LES scenes is a key topic for follow-up studies (alongside with incorporating solar radiation, etc.), as the representativeness of current results can be established only by examining further scenes.
* * *
*We added the suggested outlook to the summary section.*

Specific comments:

Page 1, Lines 9-10: The meaning of "slab-averaged applications" is not clear to me. Also, the comma after "profile" in Line 11 is not needed.
* * *
*We replaced "slab average" by " a horizontal average of the 1D and 3D radiation in each layer is used"; the comma is removed.*

Page 1, Lines 16-17: It would be important to clarify right at the first mention what is meant by "organization" and/or "organization effects" (e.g., fewer but larger clouds).
* * *
*We modified the abstract. The term 'organization' is now explained. In addition, we also refer to the differences between the 50m and 100m resolution simulations.*

Page 5, Lines 6-7: It would help to point out that averaging is over the entire scenes, including even cloud-free grid cells. (This is clarified in the last sentence of Page 13,

but readers may wonder well before that.)
* * *
*We added this information on page 5: 'These averaged heating rates are then applied in the entire layer to clear sky and cloudy regions.'*

Page 5, line 11:  I suggest deleting the sentence "The overall cooling in a modeling domain is generally stronger in case of 3D thermal NCA radiation", as the results are discussed and explained later, while this section discusses only the experimental setup.
* * *
*The sentence is deleted.*

Page 6, Line 6: I wonder in what sense does cooling compensate for the temperature perturbation.
* * *
*We summed the occurring cooling over time. After 40min, the amount of cooling brought into the system by thermal radiation is close to the temperature perturbation of each simulations (e.g. 0.8 or 1.6 K). We changed the text to: "Summing up the thermal cooling in our simulations over time, we found that 40 min is about the time it takes for the thermal cooling to compensate the original heat perturbation of the bubble. This time period is roughly the same in the strong and weakly forced case, because the stronger forced single clouds contain more liquid water and therefore more thermal cooling."*

Figure 3 and most subsequent figures: Using longer dashes in all figures would really help, as I could distinguish dashed lines from solid ones only after strong zooming. Figure 5 caption: It is not quite clear to me what "bottom" and "middle" mean in "bottom right axis" and "middle left axis."
* * *
*We modified the figures according to the suggestions.*
*The caption of Figure 5 was modified to: 'Liquid water mixing ratio is shown in pale colors on the bottom, the vertical velocity is shown in middle of each figure. The corresponding axes are on the right and left respectively'.*

Page 10, line 6:  It might be worth pointing out that relative humidity is lower in interactive simulations even though the temperature is also lower, because the liquid water mixing ratio is higher.
* * *
*We added this suggestion to the text:  'We note here that in the cloud layer, the relative humidity decreases in the local radiation simulations, because the liquid water mixing ratio is higher, although the temperature is lower.'*

Figure 10-11 captions: For clarity, I suggest replacing "as an 3 hour averaged" by "and as 3 hour averages centered" (perhaps using "starting" instead of "centered").
* * *
*We replaced the phrase according to the suggestion.*

Page 12, Lines 27-29: It seems worth mentioning that the issues of cloud organization and the size-dependence of cloud lifetime are closely related, as smaller clouds dying and larger clouds growing will result in fewer but larger clouds and in longer correlation lengths. Also, it seems worth pointing out explicitly that it is the same entrainment-invigoration due to 3D interactive radiation that reduces cloud diameter for the cylindrical cloud and erodes small-size clouds for the LES cumulus scene (if this is correct).
* * *
*We added the suggestion. We strongly suspect that interactive radiation reduces the cloud diameter, however this is not easily shown in our current simulation. This was added as a possible explanation.*

Page 13, Lines 7-8: I don't quite understand the sentence "The separation into moist and dry regions is stronger in the simulation with a coarser resolution.", and so clarification would be helpful.
* * *
*We meant to point out that in the 100m resolution simulations, we find areas covered by large clouds where most of the liquid water is located, while on the same time, cloud free areas (dry areas) exist.*
*We changed the sentence to the following which hopefully is more precise: 'We find larger areas covered by clouds and on the same time larger (drier) regions where no clouds from.'*

Page 13, Line 18: The comma after "both" can be deleted.
* * *
*The comma is deleted.*

Page 13, Lines 27-29: It seems more important to emphasize the behavior before (rather than after) 20 hours, as that is the time period for which cloud organization results are presented (Figures 13 & 14). The time after 20 hours may be mentioned in passing, but the key point is that in the first 20 hours, clouds are larger in the interactive runs.
* * *
*The conclusion is rewritten and now accounts for the suggestion.*

Page 14, Lines 17-18: The sentence "
:::
it is not certain that we would ever reach the
stage where clouds organize in the averaged radiation simulations, but we may reach the stage in the interactive ones" is confusing, because the paper discussed cloud organization in Section 3.2.2 and did not find it negligible in the interactive simulations.
* * *
*In our rewritten conclusion, this sentence is deleted.*

Page 14, Lines 3-5 also talk about significant cloud organization.
* * *
*See comment above.*

Page 14, last sentence of summary: This is a very important sentence, and even I suggest directly pointing out its main implication, that the impact of 3D effects comes from changing the spatial distribution (and not the mean value) of cooling.
* * *
*This implication is now included in the conclusion.*

Figure 19 is very helpful and I would even consider bringing it earlier, accompanied by some discussion of the key processes involved. For example, it could help to point out that the difference between 3D averaged and 3D interactive simulations is determined by the balance of two competing processes. In interactive runs, the stronger entrainment caused by cloud side cooling shrinks clouds, while the lack of cooling in the middle of updraft pockets leads to stronger updrafts and helps clouds grow. The balance of these two processes varies with the perimeter to area ratio of updrafts, and so the first process can be expected to win for small clouds, and the second one for large clouds. Finally, a minor point is that it would help to include a title for each panel or to specify in the caption what the top and bottom panels are for.
* * *
*We shifted Figure 19 to the main part of the paper (Section 3.2.2) accompanied by a summarizing text of the results. A title for each panel was added.*

Appendix: I don't think there is a need for a separate Appendix, as the current Appendix contains only the two tables that could easily be moved into the main body. Also, it would be important to clarify what is meant by "vertical stretching".
* * *
*The Appendix is removed and the contents were moved to the main text.*

---

## Author Comment (AC3) · 18 Feb 2017

**_Response to the comments of reviewer 3_**

_Our responses are marked in italic and color._

_We thank reviewer 3 for his helpful comments which helped us to improve the manuscript._

Summary:
The stated goal of this paper is to illustrate the effect of 3D thermal radiative transfer on simulated clouds. This is done by simulating a single plume cloud and a field of oceanic cumulus clouds using suite of thermal radiation configurations, ranging from no radiation to 3D thermal radiation. While this is an interesting study it is not clear what new information is brought forth. There are papers in the literature which discuss the effect of turning radiation on and off and the effect of using average instead of local radiation. New results should be the effect of interactive 3D thermal radiative transfer on the simulation but the author has missed, or at least not referenced, a key paper which has presented this sort of experiment. In addition, the results from single simulations are not particularly convincing of a clearly different response when using 3D, versus ICA, thermal radiative transfer.

With these comments in mind, I suggest that the authors perform a major revision of the paper taking into account the comments below and previous results in the literature. General comments:

Single cloud results: I would say that the results when using 1D and 3D thermal RT are almost indistinguishable and it would be challenging to read much into these results. For example, if you performed the simulations with 1D thermal RT again, or several times, but with a small random perturbation, I suspect the results would be about as different as that between the 1D and 3D. The main thing I can take from these simulations is that interactive "local" radiation has an influence versus no radiation but the complexity of the radiation parametrization doesn't seem to be too important, at least for the variables included in the analysis.
* * *
_The differences between 1D and 3D thermal radiation in this application are indeed not large. The main result is a stronger downward motion at the clouds side. However, to our knowledge, no direct comparison between the 1D and 3D thermal radiation effects on the development of a single shallow cloud exist. The study of Guan et al., 1997 only compares a no-radiation to a 3D radiation simulation. With this analysis, we tried to bridge the existing gap._
_The effects of 3D thermal radiation may very likely depend on the amount of cloud side cooling, which again depends on the cloud side area. Our simulations produce clouds that are much broader than high. Therefore only small cloud side areas exist, which might also limit the 3D effects. The performance of the NCA might be another reason why these differences are small. We included an additional subsection in the revised manuscript where we address the performance of the NCA._

Cloud field results: I have the same comments here as for the single cloud experiment. For most of the variables the differences between results for 1D and 3D "local" thermal radiation are very similar. Is there an expectation that ensembles of simulations would show the differences to be statistically significant?
* * *
*We added more discussion on the 100m resolution simulation, where 3D effects are stronger. As also commented, this is a rather surprising result, but we found that the NCA neglects some of the cloud side cooling early in the simulation in the 50m resolution simulation which might reduce the 3D effect significantly. We agree that running an ensemble would be useful to interpret results, but the simulations are quite expensive. By adding the 100m solution (which we repeated three times for the same setup) we actually increased the difference between 1D and 3D results, explaining that the 3D approximation misses part of the cloud side cooling at 50m resolution.*

Specific comments:

Title:  It does not accurately reflect of the contents of the paper.  The 3D thermal radiation is a relatively small part of the paper and the paper focuses on a very particular types of cloud. I.e, I don't think the results could be generalized to all clouds.
* * *
*We changed the title to: 'Effects of 3D Thermal Radiation on the Development of a Shallow Cumulus Cloud Field'*

References: The highly relevant paper by Mechem is missing:
Mechem, D. B.; Kogan, Y. L.; Ovtchinnikov, M.; Davis, A. B.; Evans, K. F. & Ellingson, R. G. Multidimensional Longwave Forcing of Boundary Layer Cloud Systems Journal of the Atmospheric Sciences, 2008, 65, 3963-3977.
There are some papers that have discussed interactive cloud resolving simulations considering aspects of 3D solar radiative transfer:

Koracin, D.; Isakov, V. & Mendez-Nunez, L. A cloud-resolving model with the radiation scheme based on the Monte Carlo method Atmospheric Research, 1998, 47-48, 437-459
Frame,  J. & Markowski,  P. Numerical Simulations of Radiative Cooling beneath the Anvils of Supercell Thunderstorms Monthly Weather Review, 2010, 138, 3024-3047

And there are papers that discuss the effect of using domain mean radiative fluxes (here I give just two examples, I am sure there are others),
Petch, J. C. and Gray, M. E. B. (2001), Sensitivity studies using a cloud-resolving model simulation of the tropical west Pacific. Q.J.R. Meteorol. Soc., 127: 2287–2306. doi:10.1002/qj.49712757705
Cole, J. N. S.; Barker, H. W.; Randall, D. A.; Khairoutdinov, M. F. & Clothiaux, E. E. Global consequences of interactions between clouds and radiation at scales unresolved by global climate models Geophysical Research Letters, 2005, 32, L06703
* * *
*We added the missing papers and even more to the introduction and the discussion of the results.*

Introduction, long paragraph stating at line 3, page 2: This paragraph is challenging to read and needs to rewritten since in its current state it comes across as an "information dump".  From it reader needs to pull together information needed for the remainder of the paper.  Breaking the paragraph into at least two would help as would putting the information into an order that fits with rest of the paper.  I.e., general effect of thermal

radiation on cloud development, previous results 1D versus 3D thermal radiation and a justification for examining local versus non-local radiation (1D versus slab averages) with discussion of previous results.
* * *
*We restructured the introduction according to the suggestion and added the suggested literature.*

Page 2 line 14:  The discussion of the Guan study is a bit unclear since that study compared 3D thermal radiation against the case of no radiation, not versus 1D radiative transfer,
* * *
*The Guan study is discussed in more detail in the rewritten introduction.*

Page 3, paragraph at line 5: Did you add modify the equations used for the cloud
microphysics to explicitly model enhanced emission by drops?  My understanding of the papers by Harrington is that a term for the radiative heating and cooling of the drop is considered. If you did not add drop cooling to the microphysics the interpretation of this paragraph is tricky.
* * *
*We did not modify the microphysics parametrization. We added in advance of this paragraph in the introduction the following sentence:*
*'The microphysical aspect mentioned before will not be addressed in this study, however, for the matter of completenes, we will briefly point out what was found in the past:'*

Page 4, line 23:  Is the shape of the cloud that sensitive to the structure of the pertur-bation?
* * *
*Not necessarily, however, by chance the random numbers could be distributed in a way that produces a very different cloud. To avoid that we used the same random numbers to make sure that we really apply the radiative transfer to the same initial cloud.*

Page 5, line 27: Why not show the clouds simulated using the 3D thermal RT? Would it not be the most realistic? Also showing the cloud field at the 20 minute point does not make sense given the discussion in the text.  The text it is pointed out that it is cloud field in the period 40-80 minutes that are to be the focus of the analysis. Why not show the cloud at that point?
* * *
*As this figure was only meant to give the reader an idea about the shape of the cloud, we chose an early time step where the clouds are still pretty similar. But we changed the figure according to the suggestion.*

Page 6, line 13:  The meaning of this statement 'from about 30 min onward the cloud stays rather constant at a certain height' is not clear. What exactly stays constant (liquid water path?)?
* * *
*What we meant is that the cloud stays in one height and does not rise further
up as it has until this time step. We changed the sentence as follows:
'... 30 min onward the cloud stays rather constant at a certain height and does not rise
any further ...'*

Page 6, line 15: This sentence,
"All simulations show that the liquid water path (top row of Fig. 3) is reduced by thermal
radiation in this "second stage" (from about 20 min to 40 min)."
is not clear. The same reduction is seen in all simulations without any radiation. Do you
mean to say that thermal radiation causes liquid water path to be less than the case
with no radiation? This difference is pretty small.
* * *
*We deleted 'thermal radiation' in the sentence*

Page 7,  lines 1-27:  This analysis seems to end abruptly or I'm missing something.
There is an idea of "subsiding shells" to explain the subsidence around the edge of the
cloud.  As mentioned in the text, Heus and Jonker, 2005 attribute the presence of the
shell to "negative buoyancy, resulting from evaporative cooling following lateral mixing
of environmental air with cloudy air.". The results with thermal radiation have downdraft
shells that are stronger than that in the no-radiation case.
Is it not possible to use the output from the model to further analyze and show why
the shell is enhanced in the presence of thermal radiation?  Is it the radiation directly
producing more negative buoyancy, Figure 1 suggests large radiative cooling, or does
it induce an environment that enhances the evaporative cooling? It must be possible to
quantify statements like "This might be due to the thermal cooling at cloud tops, and in
case of 3D Thermal NCA radiation at cloud sides." and "The stronger horizontal buoy-
ancy gradient (difference between positive and negative buoyancy in Fig 6) generates
enhanced turbulence and therefore stronger evaporation.".
* * *
*Figure 6 shows the profiles of positive and negative buoyancy. This figure is
useful in two ways. First, it shows that both thermal radiation simulations
have larger values of negative buoyancy, which, as pointed out by Heus and
Jonker, leads to the development of the subsiding shell. It is therefore the
direct cooling of the thermal radiation which initiates the subsiding shell at
first.
Second, this figure shows the horizontal buoyancy gradient which can be used
to indicate evaporation at the cloud side. As the horizontal buoyancy gradient
is stronger in case of the thermal radiation simulations, more evaporation is
occurring, reducing the diameter of the cloud as seen in Figure 5. At the same
time, there is of course more evaporative cooling produces. With our current
simulations, it is not possible to differentiate at this point, if thermal
radiation or evaporative cooling is driving the subsiding shell any
further. But thermal radiation certainly initiates the development.
We modified this passage and hope that it is more precise now.*

Section 3.2:  What is special about the "restart time" at 3 hours?  Was the model run
differently up to this point?
* * *
*We chose the restart time at 3h, as it is the time after spin up and where the*

*first clouds occur. For all simulations, the model run is therefore the same until this point. The initial simulation was driven by 1D solar and thermal radiation. From 3 hours on, we apply the different radiation setups. We modified our description in the model setup (section 2.1) as follows:*
*'All simulations are restarted and analyzed after a 3~h initialization run. Until 3~h, the initial simulation is driven by 1D solar and thermal radiation. From the restart time on, we switch on one of the five thermal radiation application or switch radiation off, thus skipping the spin-up. At 3~h, the first clouds form in the initial run.'*
*We show the figures from this time on, to see the development from the same initial state of all simulations.*

Page 8, line 15: Are the liquid water path and other variables shown for this case averaged over the entire domain or sampled only over clouds?
* * *
*Domain variables, if not otherwise stated are sampled in the entire domain. We added this information to the text.*

Page 8, line 17: How robust are these results?
* * *
*The differences do indeed not seem to be large. We would love to repeat he simulations various times and look at the statistical significance, however this was due to computational cost and storage space not possible. We repeated the simulation on 100m resolution and find (in 3 simulations there) the same behavior. It is clear however that 3 simulations are still not enough to provide statistical evidence. From other variables it is however clear that there is a definite difference between averaged and local radiation. For the liquid water path this might simply be smaller, as the cloud cover is higher in the averaged radiation simulations, but more liquid water is found in individual clouds (see e.g. max. liquid water content) in the local radiation simulations. This might lead to the small differences in this case. We added to this sentence:*
*'This differences are small however and might be a result of the larger cloud fraction but less maximum liquid water of the averaged radiation simulations versus the reduced cloud cover and higher maximum liquid water content in the local thermal radiation simulations.'*

Page 8, line 19: Is it an expected result that "All quantities increase over time.". If you continued running the simulation would it go into a quasi-equilibrium state?
* * *
*It is in some way expected as we add more and more cooling to the system by the thermal radiation, but do not allow for rain. Therefore more and more water should condense, cloud cover should increase and at some point (if we would drive the simulations longer), we would get a cloud cover of 100%.*

Page 8, line 20: Remove this sentence as it is obvious,
"The different development of the No-Radiation simulation and the radiation simulations is related to the missing cooling of the thermal radiation in the No-Radiation simulation."
* * *
*The sentence is removed.*

Page 8, line 23: Why does the lack of thermal radiative cooling lead to a higher cloud base? It is not clear to me.
* * *
*Simply by a shift of the condensation level. The temperature profile is shifted by 1-2 degree to higher values in the no-radiation simulation. Therefore condensation will take place further up in the atmosphere where it is cooler.*
*We modified the sentence as follows:*
*'The higher cloud base is also a result of the missing cooling which leads to a warmer temperature profile in the No-Radiation simulation'*

Page 8, line 25:  I don't think you want to use the term "bias" here, perhaps the word "change" instead?
* * *
*We changed the word to 'change'.*

Page 8, line 26; Perhaps a clear term than "interactive" would be "local" since the
"averaged" radiation is also interactive since it still reacts to changes in the clouds.
* * *
*We changed 'interactive' to 'local'.*

Page 8, line 29: What is so interesting about the liquid water path and maximum liquid water content?
* * *
*It is interesting that the liquid water path and maximum liquid water is lower, while the cloud cover is higher in the averaged radiation simulations. We changed the sentence to:*
*'Liquid water path and maximum liquid water content develop in the opposite direction: both are lower for the averaged radiation simulations until 20 hours'*

Page 9, line 4: The rate of increase in cloud fraction for the "interactive" radiation after hour 22 is nearly as large as for the averaged radiation. How does this fit into the organization hypothesis?
* * *
*This sentence is removed in the revised manuscript. Earlier in the text it is revered to development of the cloud field after 20 hours in the following way:*
*'It shall be mentioned here (although not shown) that from about 24 hours on, large clouds form in the averaged radiation simulations and the const cooling simulation, in which the clouds oscillate: disappearing and then reappearing. No systematic difference between 1D and 3D radiation is found in these cases. The local radiation simulations still show cells, however, clouds become larger, especially in the 3D Thermal NCA simulation.'*

Page 9, line 13: Initial profiles in first column not first row.
* * *
*We changed the text accordingly.*

Page 9, line 26: How significant is the approximately 5% greater liquid water content in the 3D NCA simulation at hour 10?
* * *
*As pointed out before, it was not possible to re-run the simulation various times. The differences are indeed not large at this point in time but a tendency of what is happening in the following can already be seen. The liquid water path at 20h shows already a difference of 10%.*

Page 9, line 33: Figure 11 first column, not Figure 11 (second column)?
* * *
*We refer indeed to Figure 11, second column. The sentence before relates to the increased buoyancy production at 10hrs, which is shown in the second column.*

Page 10, line 1: The 3D NCA simulations produce slightly more TKE through buoyancy in the upper cloud with stronger upward and downward vertical winds. Again, is this significant? As discussed further down in this section the more significant result is that horizontal averaging causes more significant differences.
* * *
*We fully agree that there is more difference between the averaged and the local radiation simulations then between 1D and 3D thermal radiation simulations. However, we find stronger 3D effects in the 100m resolution simulation and explain this in the revised paper. Essentially, in the 50m resolution simulation the NCA misses some of the cloud side cooling, wherefore the 3D effects are only very small.*

Page 10, line 24: From the results shown I would suggest that it is not conclusive that "3D Thermal NCA" increases the results shown. The differences relative to 1D ICA are quite modest and it is not clear if they are by chance or systematic.
* * *
*We changed the sentence to:*
*'3D Thermal NCA radiation, in comparison to 1D thermal radiation shows a slightly stronger increase of these shown effects by an additional cloud side cooling and overall stronger cooling in the modeling domain.'*

*In general, we tried to point out throughout the text of the revised manuscript that 3D effects seen in these statistical variables show usually slightly higher values, but only slightly.*

Page 11, line 2: spacial -> spatial
* * *
*We changed the text accordingly.*

Page 12, line 14: Can you quantify that "we find larger structures earlier in the 3D Thermal NCA radiation simulation"? Staring at the plots for 1D ICA and 3D NCA in Figure 14, it is not clear how one objectively comes to this conclusion. The color contouring gives some bias toward clouds with larger liquid water path, not necessarily

larger cloud structures.
* * *
*We added an additional figure of the cloud field which hows the spatial distribution of the clouds.*
*The above conclusion comes from a combination of the autocorrelation, looking into the data and the hovemoeller diagrams. We have rewritten this paragraph and hope it is more clear now.*

Section 3.2.3: Were 3 simulations performed for each radiation configuration? If so, did the "small differences" lead to a stronger or weaker case for differences between the 1D and 3D interactive simulations? The results of the simulations, especially 3D NCA, seems rather sensitive to horizontal resolution (Figs. 17 and 18). Any speculation as to why? Should we expect different results if we reduced the horizontal resolution to 25 m?
* * *
*We added the additional two runs in two figures in the revised draft to show the differences in the simulations. The new and additional chapter about the performance of the NCA shows why we see a stronger 3D effect in the 100m resolution simulation. Increasing the resolution to 25m is with the NCA not appropriate at present. It would require further development of the parametrization. Following our conclusions here, we would get 3D effects at 25m resolution with an improved parameterization.*

Page 14, line 25: It is not a strong or clear result in this paper that the 3D interactive radiation is significantly stronger than the 1D ICA radiative transfer. Therefore, this statement is not well supported.
* * *
*The conclusion is rewritten.*

Table 1: Horizontal resolution does not match with text, 100 m in table and 50 m in text. If it is the latter then number of gridboxes is incorrect since domain size is quoted in text to 6.4 km by 6.4 km.
* * *
*The tables are moved to the main part of the text as suggested by reviewer 2 and are consistent with the text now.*

Figure 5: It is very difficult to read this figure. For example, the dashed lines are almost impossible to see on the printed document. For this figure titles indicating which are symmetric cloud and which are non-symmetric clouds is warranted.
* * *
*We modified the figures accordingly.*

Figures 10 and 11: The solid versus dash in the legend is very difficult to make out.
* * *
*We modified the figures accordingly.*

---

## Author Response (AR2)

**Response to the comments of the editor:**

We thank the editor for her advice.

We addressed the point raised by reviewer 3. We prefer not exclude the discussion of the 50m horizontal resolution simulation, because we think that it adds important information. Our reasons are given in the response to the reviewer below. Spelling mistakes have been removed. The detailed literature overview is mainly due to the reviewer requests for a more comprehensive literature discussion.

**Response to the comments of reviewer 1:**

We thank reviewer 3 for his continuous effort in improving our paper.

We prefer to keep both the 50m and 100m horizontal resolution simulation in our paper, because both contribute complimentary information:
Showing the simulations of both resolutions is a compromise of the optimal use of the NCA and a better resolution for boundary layer structures. The small scale structures (e.g. the subsiding shell) are better resolved in the 50m resolution simulation and although some part of the 3D cooling is missing, the effect on the subsiding shell can be seen. The simulations at 100m horizontal resolution show in addition stronger feedback of the 3D thermal radiation, namely the cloud organization, because the 3D thermal cooling is on average better represented.

We added our thoughts on this decision in the paper as follows:

'Although some of the 3D cooling is missing in the beginning of the 50m resolution simulation, we decided to show the 50m and the 100m resolution simulation in comparison. While the 100m resolution simulation represents the 3D cooling (on average) better, the 50m resolution simulations resolves small scale structures (as e.g. the subsiding shell) better. '

The paper has been read another time and spelling mistakes including those mentioned by the reviewer have been corrected.

**Response to the comments of reviewer 2:**

The paper has been read another time and spelling mistakes have been corrected.

[revised manuscript text omitted]